



**Factors controlling plankton productivity, particulate matter stoichiometry, and export**
**flux in the coastal upwelling system off Peru**
Lennart Thomas Bach[1]*, Allanah Joy Paul[2], Tim Boxhammer[2], Elisabeth von der Esch[3],
Michelle Graco[4], Kai Georg Schulz[5], Eric Achterberg[2], Paulina Aguayo[6], Javier Arístegui[7],
Patrizia Ayón[4], Isabel Baños[7], Avy Bernales[4], Anne Sophie Boegeholz[8], Francisco Chavez[9],
Shao-Min Chen[2,10], Kristin Doering[2,10], Alba Filella[2], Martin Fischer[8], Patrizia Grasse[2], Mathias
Haunost[2], Jan Hennke[2], Nauzet Hernández-Hernández[7], Mark Hopwood[2], Maricarmen
Igarza[11], Verena Kalter[2,12], Leila Kittu[2], Peter Kohnert[2], Jesus Ledesma[4], Christian Lieberum[2],
Silke Lischka[2], Carolin Löscher[13], Andrea Ludwig[2], Ursula Mendoza[4], Jana Meyer[2], Judith
Meyer[2], Fabrizio Minutolo[2], Joaquin Ortiz Cortes[2], Jonna Piiparinen[12], Claudia Sforna[2],
Kristian Spilling[14,15], Sonia Sanchez,[4] Carsten Spisla[2], Michael Sswat[2], Mabel Zavala
Moreira[16], Ulf Riebesell[2]
*corresponding author: Lennart.bach@utas.edu.au
[1]Institute for Marine and Antarctic Studies, University of Tasmania, Hobart, Tasmania, Australia
[2]GEOMAR Helmholtz Centre for Ocean Research Kiel, Kiel, Germany
[3]Institute of Hydrochemistry, Chair of Analytical Chemistry and Water Chemistry, Technical University
of Munich, Munich, Germany
[4]Dirección General de Investigaciones Oceanográficas y cambio Climático, Instituto del Mar del Perú
(IMARPE), Callao, Perú
[5]Centre for Coastal Biogeochemistry, School of Environment, Science and Engineering, Southern
Cross University, Lismore, NSW, Australia
[6]Millennium Institute of Oceanography (IMO), Universidad de Concepción, Concepción, Chile
[7]Instituto de Oceanografía y Cambio Global, IOCAG, Universidad de Las Palmas de Gran Canaria
ULPGC, Las Palmas, Spain

25● [8]Department of Biology, Institute for General Microbiology, Christian-Albrechts-Universität zu Kiel,

Kiel, Germany
[9]Monterey Bay Aquarium Research Institute, Moss Landing, United States of America



[10]Department of Earth Sciences, Dalhousie University, Halifax, Canada

29•    [11]Programa de Maestría en Ciencias del Mar, Universidad Peruana Cayetano Heredia, Lima, Perú

30•    [12]Memorial University of Newfoundland, Department of Ocean Sciences, Logy Bay,
Newfoundland, Canada

32•    [13]University of Southern Denmark, Odense, Denmark

[14]Finnish Environment Institute, Marine Research Centre, Helsinki, Finland
[15]Faculty of Engineering and Science, University of Agder, Kristiansand, Norway
[16]Escuela Superior Politécnica del Litoral, Guayaquil, Ecuador
**Abstract**
Eastern boundary upwelling systems (EBUS) are among the most productive marine
ecosystems on Earth. The high productivity in surface waters is facilitated by upwelling of
nutrient-rich deep waters, with high light availability enabling fast phytoplankton growth and
nutrient utilization. However, there are numerous biotic and abiotic factors modifying
productivity and biogeochemical processes. Determining these factors is important because
EBUS are considered hotspots of climate change, and reliable predictions on their future
functioning requires understanding of the mechanisms driving biogeochemical cycles therein.
In this study, we used *in situ* mesocosms to obtain mechanistic understanding of processes
controlling productivity, organic matter export, and particulate matter stoichiometry in the
coastal Peruvian upwelling system. Therefore, eight mesocosm units with a volume of ~50 $m^3$
were deployed for 50 days ~6 km off Callao during austral summer 2017, coinciding with a
coastal El Niño event. To compare how upwelling of different water bodies influences plankton
succession patterns, we collected two subsurface waters at different locations in the regional
oxygen minimum zone (OMZ) and injected these into four replicate mesocosms, respectively
(mixing ratio ≈ 1.5:1 mesocosm: OMZ water). The differences in nutrient concentrations
between the collected water bodies were relatively small, and therefore we do not consider
treatment differences in the present paper. The phytoplankton communities were initially
dominated by diatoms but shifted towards a pronounced dominance of the mixotrophic harmful
dinoflagellate (*Akashiwo sanguinea*) when inorganic nitrogen was exhausted in surface layers.
The community shift resulted in a major short-term increase in productivity during *A. sanguinea*





growth which left a pronounced imprint on organic matter C:N:P stoichiometry. However, C,
N, and P export fluxes were not affected by this ecological regime shift because *A. sanguinea*
persisted in the water column and did not sink out during the experiment. Accordingly, ongoing
export fluxes during the study were maintained mainly by a remaining "background" plankton
community. Overall, biogeochemical pools and fluxes were surprisingly constant in between
the ecological regime shifts. We explain this constancy by light limitation through self-shading
by phytoplankton and inorganic nitrogen limitation which constrained phytoplankton growth.
Thus, gain and loss processes seemed to be relatively well balanced and there was little
opportunity for blooms, which represents an event where the system becomes unbalanced. The
mesocosm study revealed key links between ecological and biogeochemical processes for one
of the economically most important regions in the oceans.
**1. Introduction**
Eastern boundary upwelling systems (EBUS) are hotspots of marine life (Chavez and Messié,
2009). They support around 5 % of global ocean primary production and 20 % of marine fish
catch whilst covering less than 1 % of the ocean surface area (Carr, 2002; Chavez and Messié,
2009; Messié and Chavez, 2015).
One of the most productive EBUS is located along the Peruvian coastline between 4°S and 16°S
(Chavez and Messié, 2009). Here, southerly trade winds drive upward Ekman pumping and
offshore Ekman transport, resulting in upwelling of nutrient rich deep waters (Albert et al.,
2010). In the surface ocean, the nutrient rich water is exposed to high levels of irradiance
leading to enhanced primary production (Chavez et al., 2008).
The high primary production has two major consequences. First, large amounts of organic
material sink into subsurface water layers where they are remineralized and consume dissolved
oxygen ($O_2$). In the Pacific, these subsurface water masses are old and already depleted in $O_2$,
and the additional subsurface $O_2$ consumption in the Peruvian upwelling results in exhaustion
of the remaining oxygen leading to  one of the most pronounced oxygen minimum zones (OMZ)
of the  global ocean (Karstensen et al., 2008; Stramma et al., 2008). Second, the high primary
production fuels secondary production and sustains one of the largest fisheries in the world
which makes the Peruvian upwelling an area of outstanding economic value (Chavez et al.,
86   2008).





Highest primary production occurs near the coast from where the water is advected offshore
(Carr, 2002). While primary production is modified further offshore by eddies and other
features, it generally declines with increasing distance from shore (Chavez and Messié, 2009;
Stramma et al., 2013). Therefore, the simplified view is that plankton succession starts in the
freshly upwelled water masses near shore and continues while the waters travel offshore.
Plankton community composition changes continuously along this path. Diatom-dominated
phytoplankton and herbivore mesozooplankton often prevail near the coast, but the community
transitions towards Crypto-, Hapto-, Prasino-, and Cyanophyceae as well as a more carnivorous
mesozooplankton community further offshore (Ayón et al., 2008; DiTullio et al., 2005; Franz
et al., 2012a; Meyer et al., 2017). Also dinoflagellates can play an important role, especially
when upwelling relaxes and nutrient concentrations decrease (Smayda and Trainer, 2010). Key
biogeochemical processes such as productivity and export are closely coupled to the structure
of plankton communities (Bach et al., 2019; Boyd and Newton, 1999; Longhurst, 1995). Thus,
the observed patterns of productivity and export in the Peruvian upwelling system (and
elsewhere) can only be understood when the associated links to the plankton community
structures are revealed. Establishing and quantifying these links is particularly important for the
Peruvian upwelling system considering that this region is disproportionally affected by climate
change, and alterations in productivity could disrupt one of the largest fisheries in the world
(Gruber, 2011).
In austral summer 2017 (coincidently during a strong coastal El Niño), we set up an *in situ*
mesocosm experiment in the coastal Peruvian upwelling system near Callao to gain mechanistic
understanding on how the biological processes in the plankton community influence
biogeochemical processes. Our two primary questions were: 1) How do plankton community
structure and associated biogeochemical processes change following an upwelling event. This
first question was addressed by simply monitoring the developments within the mesocosms for
a 50 days period. 2) How does upwelling of water masses with different OMZ-signatures
influence plankton succession and pelagic biogeochemistry. This second question was
addressed by adding two different OMZ-influenced subsurface waters to 4 mesocosms,
respectively. In the present paper we will focus on the first question and target three ecologically
and biogeochemically important measures: productivity, export, and organic matter
stoichiometry. Our paper kicks off a Biogeosciences special issue for the 2017 Peru mesocosm
campaign and therefore includes a comprehensive description of the basic setup, the major
caveats, and the key results of this study.





## 2. Methods

### 2.1 Mesocosm deployment and maintenance

On February 22, 2017, eight "Kiel Off-Shore Mesocosms for Future Ocean Simulations" (KOSMOS, M1 – M8 (Riebesell et al., 2013)) were deployed with *Buque Armada Peruana (BAP) Morales* in the SE Pacific, 6 km off the Peruvian coastline (12.0555°S; 77.2348°W; Fig. 1). The water depth at the deployment site was ~30 m and the area was protected from southern to southwestern swells by Isla San Lorenzo (Fig. 1). The mesocosms consisted of cylindrical 18.7 m long polyurethane bags (2 m diameter, 54.4 ±1.3 m$^3$ volume) suspended in 8 m tall flotation frames (Fig. 1). The bags were initially folded so that the flotation frames with the packed bags could be lifted with the crane from *BAP Morales* into the water where the mesocosms were connected to anchor weights. Bags were unfolded immediately after deployment with the lower end extending to ~19.7 m and the upper end 1 m below surface. Nets (mesh size 3 mm) attached to both ends of the bags allowed water exchange but prevented larger and typically more patchily distributed plankton or nekton to enter the mesocosms. On February 25, the meshes attached to the lower end were replaced by divers with 2 m long conical sediment traps thereby sealing the bottom of the mesocosms. The upper ends of the bags were pulled ~1.5 m above surface immediately after sediment trap attachment. These two steps isolated the water mass enclosed inside the mesocosms from the surrounding Pacific and marked the beginning of the experiment (day 0, Fig. 2). Ultimately, the enclosed water columns were ~19 m deep of which the lowest 2 m were the conical sediment traps (Fig. 1).

The mesocosm bags were regularly cleaned from the inside and outside to minimize wall growth (Fig. 2). Cleaning the outside of the bags was done with brushes, either from small boats (0 – 1.5 m) or by divers (1.5 – 8 m). The inner sides of the bags were cleaned with rubber blades attached to a polyethylene ring which had the same diameter as the mesocosm bags and was ballasted with a 30 kg weight (Riebesell et al., 2013). The rubber blades were pushed against the walls by the ring and scraped off the organic material while sliding downwards. Cleaning inside down to ~1 m above the sediment traps was conducted approximately every eighth day to prevent biofouling at an early stage of its progression.

### 2.2 OMZ water addition to the mesocosms

On March 1 and 2, 2017 (day 4 and 5), we collected two batches (100 m$^3$ each) of OMZ water with Research Vessel IMARPE IV at two different stations of the IMARPE time-series transect





(Graco et al., 2017). The first batch was collected on day 5 at station 1 (12.028323°S;
77.223603°W) at a depth of 30 m. The second was collected on day 10 at station 3
(12.044333°S; 77.377583°W) at a depth of 90 m (Fig. 1). In both cases we used deep water
collectors described by Taucher et al. (2017). The pear-shaped 100 m³ bags of the collector
systems consisted of flexible fiber-reinforced food-grade polyvinyl chloride material (opaque).
The round openings of the bags (0.25 m diameter covered with a 10 mm mesh) were equipped
with a custom-made propeller system that pumped water into the bag and a shutter system that
closed the bag when full. Prior to their deployment, the bags were ballasted with a 300 kg
weight so that the bag sank to the desired depth. A rope attached to the bag guarenteed that it
did not sink deeper. The propeller and the shutter system were time-controlled and started to
fill the bag after it had reached the desired depth and closed the bag after ~1.5 hours of pumping.
To recover the collector, the weight was released with an acoustic trigger so that 24 small floats
attached to the top made the system positively buoyant and brought it back to the surface. The
collectors were towed back to the mesocosm area and moored therein with anchor weights.
On March 8 and 9, 2017 (day 11 and 12), we exchanged ~20 m³ of water enclosed in each
mesocosm with water collected from station 3 (M2, M3, M6, M7) or station 1 (M1, M4, M5,
M8). The exchange was done in two steps using a submersible pump (Grundfos SP-17-5R,
pump rate ~18 m³ h⁻¹). On day 8, we installed the pump for about 30 – 40 minutes in each
mesocosm and pumped 9 m³ out of each bag from a depth of 11 – 12 m. On day 11, the pump
was installed inside the collector bags and 10 m³ of water was injected to 14 – 17m depth (hose
diameter 5 cm). Please note that the pump (for water withdrawal) and hose (for water injection)
were carefully moved up and down the water column between 14 – 17 m so that the water was
evenly withdrawn from or injected into this depth range. On day 12, we repeated this entire
procedure but this time removed 10 m³ from 8 – 9 m, and added 12 m³ evenly to the depth range
from 1 – 9 m.
**2.3 Salt additions to control stratification and to determine mesocosm volumes**
Oxygen minimum zones are a significant feature of EBUS and play an important role for
ecological and biogeochemical processes in the Humboldt system. They reach very close to the
surface (<10 m) in the near-coast region of Peru (Graco et al., 2017), thus leading to an inclusion
of water masses with low bottom $O_2$ concentrations in the mesocosms below ~10 m (see
results). Conserving the low $O_2$ bottom layer within the mesocosms throughout the experiment
required an artificial water column stratification because otherwise heat exchange with the





surrounding Pacific water would have induced significant convective mixing which would have
destroyed this feature (see Bach et al. 2016 for a thorough description of the convective mixing
phenomenon in mesocosms). Therefore, we injected 69 L of a concentrated NaCl brine solution
evenly into the bottom layers of the mesocosms on day 13 by carefully moving a custom made
distribution device (Riebesell et al., 2013) up and down between 10 – 17 m. The procedure was
repeated on day 33 with 46 L NaCl brine solution added between 12.5 – 17 m which was
necessary because turbulent mixing between days 13 and 33 continuously blurred the artificial
halocline. The brine additions increased bottom water salinity by about 1 during both additions
(Fig. 3B).
At the end of the experiment (day 50; after the last sampling), we performed a third NaCl brine
addition but this time with the purpose to determine the volume of each mesocosm. For volume
determination, we first homogenized the enclosed water columns by pumping compressed air
into the bottom layer for 5 minutes, thereby fully mixing the water masses. This was validated
by salinity profiling with subsequent CTD casts (see section 2.4 for CTD specifications). Next,
we added 52 kg of a NaCl brine evenly to the entire water column as described above, followed
by a second airlift mixing and second set of CTD casts. Since we precisely knew the added
amount of NaCl, we were able to determine the volume of the mesocosms at day 50 from the
measured salinity increase as described by Czerny et al. (2013). The mesocosm volumes before
day 50 were calculated for each sampling day based on the amount volume that was withdrawn
during sampling (section 2.5) and exchanged during the OMZ water addition (section 2.2).
Rainfall did not occur during the study and evaporation was negligible ($\sim$1 L d$^{-1}$) as determined
by monitoring salinity over time (section 2.5). These two factors were therefore not considered
for the volume calculations.
The NaCl solution used to establish haloclines was prepared in Germany by dissolving 300 kg
of food industry grade NaCl free of anti-caking agents in 1000 L deionized water (Milli-Q,
Millipore) (Czerny et al., 2013). The brine was purified thereafter with ion exchange resin
(Lewawit$^{TM}$ MonoPlus TP260$^®$, Lanxess, Germany) to minimize potential contaminations with
trace metals (Czerny et al., 2013). Therefore, the NaCl dissolved in deionized  water was
pumped through acid cleaned columns which contained the ion exchange granulate. The
purified brine was collected in an acid cleaned polyethylene canister (1000 L), sealed, and
transported from Germany to Peru where it was used $\sim$5 months later. The brine solution for
the volume determination at the end of the experiment was produced on site using table salt
purchased locally.



### 2.4 Additions of organisms

Some of the research questions of this campaign involved endemic organisms that were initially not enclosed in the mesocosms, at least not in sufficient quantities for meaningful quantitative analyses. These were scallop larvae (*Argopecten purpuratus*, "Peruvian scallop") and eggs of the fish *Paralichthys adspersus* ("Fine flounder"). Both scallop larvae and fish eggs were introduced by lowering a container with the organisms to the water surface and carefully releasing them into the mesocosms. Scallop larvae were added on day 14 in concentrations of ~10.000 individuals $m^{-3}$. Fish eggs were added on day 31 in concentrations of ~90 individuals $m^{-3}$. However, few scallop larvae and no fish larvae were found in the mesocosms after the release so that their influence on the plankton community should have been small and will only be considered in specific zooplankton papers in this special issue.

### 2.5 Sampling and CTD casts

Sampling and CTD casts were undertaken from small boats that departed the harbor in La Punta (Callao, Fig. 1) around 6.30 a.m. (local time) and reached the study site around 7 a.m. The sampling scheme was consistent throughout the study, starting with the sediment traps to avoid resuspension of the settled material during deployment of our sampling gear. This was followed by water column sampling and CTD casts, starting ~10 minutes after sediment trap sampling. The entire sediment trap sampling lasted for one hour while the CTD casts lasted for 2 hours after which both sampling teams went back to the harbor. Water column sampling for all parameters except mesozooplankton lasted for 2 – 6 hours (mostly 3 hours) and the crew arrived back in the harbor mostly between 11 a.m. and 2 p.m. Mesozooplankton was sampled in the afternoon between 1 – 5 p.m using and Apstein net of 17 cm diameter and 100 µm mesh size (Lischka et al., 2017). Care was taken to sample mesocosms and Pacific surface waters (which was sampled alongside the mesocosms during every sampling) in random order. Sampling containers were stored in cool boxes until further processing on land. Details of the individual sampling procedures are described in the following where necessary.

Sinking detritus was collected in the sediment traps at the bottom of each mesocosm and recovered from there every second day (Fig. 2) with the vacuum pumping system described by Boxhammer et al. (2016). Briefly, a silicon hose (10 mm inner diameter) attached to the collector at the very bottom of the traps led to the surface where it was fixed above sea level at one of the pylons of the flotation frame and closed with a clip (Fig. 1A). The sampling crew attached a 5 L glass bottle (Schott Duran) to the upper end of the hose and generated a vacuum



(~300 mbar) within the bottle using a manual kitesurf pump so that the sediment material was
sucked through the hose and collected in the 5 L bottle after the clip was loosened.
Suspended and dissolved substances investigated in this study comprised particulate organic
carbon (POC) and nitrogen (PON), total particulate carbon (TPC) and phosphorus (TPP),
biogenic silica (BSi), phytoplankton pigments, nitrate ($NO_3^-$), nitrite ($NO_2^-$), phosphate ($PO_4^{3-}$
), silicic acid ($Si(OH)_4$), ammonium ($NH_4^+$), dissolved organic nitrogen (DON) and phosphorus
(DOP). Suspended and dissolved substances were collected with 5 L "integrating water
samplers (IWS)" (Hydro-Bios Kiel) which are equipped with pressure sensors to collect water
evenly within a desired depth range. We sampled two separate depth ranges (surface and bottom
water). These depth ranges were 0 – 5 and 5 – 17 m from day 1 to 2, 0 – 10 and 10 – 17 m from
day 3 to 28, and 0 – 12.5 and 12.5 – 17 m from day 29 to 50 (Fig. 2). The reason for this
separation was that we wanted to have specific samples for the low $O_2$ bottom water. However,
for the present paper we only show IWS-collected data averaged over the entire water column
(0 – 17 m) as this was more appropriate for the data evaluation within this particular paper (for
example; POC on day 30 = (12.5 * $POC_{0-12.5m}$ + 4.5 * $POC_{12.5-17m}$) / 17). Surface and bottom
water for POC, PON, TPC, TPP, BSi, and phytoplankton pigments were carefully transferred
from the IWS into separate 10 L polyethylene carboys. Samples for inorganic and organic
nutrients were transferred into 250 mL polypropylene and acid-cleaned glass bottles,
respectively. All containers were rinsed with Milli-Q water in the laboratory and pre-rinsed
with sample water immediately before transferring the actual samples. Trace metal clean
sampling was restricted to 3 occasions (days 3, 17 and 48) due to logistical constraints.
Therefore, acid-cleaned plastic tubing was fitted to a Teflon pump, submerged directly into the
mesocosms and used to pump water from surface and bottom waters (depths as per
macronutrients) for the collection of water under trace-metal clean conditions.
Depth profiles of salinity, temperature, $O_2$ concentration, photosynthetically active radiation
(PAR), and chlorophyll a (chl-a) fluorescence were measured with vertical casts of a CTD60M
sensor system (Sea & Sun Technologies) on each sampling day (Fig. 2). Details of the salinity,
temperature, PAR, and fluorescence sensors were described by Schulz and Riebesell (2013).
The Fast Oxygen Optical Sensor measured dissolved $O_2$ concentrations at 620 nm excitation
and 760 nm detection wavelengths. The sensor is equipped with a separate temperature sensor
for internal calculation and linearization. It has a response time of 2 s and was calibrated with
$O_2$ saturated and $O_2$ deplete seawater. Absolute concentrations at discrete depths were
compared with Winkler $O_2$ titration measurements. These were taken in triplicate with a Niskin



sampler on day 40 at 15 m water depth in M8 and on day 42 at 1 m in M3. Samples were filled
into glass bottles allowing significant overflow and closed air-tight without headspace. All
samples were measured on the same day with a Micro Winkler titration device as described by
Arístegui and Harrison (2002). We only used CTD data from the downward cast since the
instrument has no pump to supply the sensors mounted at the bottom with a constant water
flow. A 3 min latency period with the CTD hanging at ~2 m before the casts ensured sensor
acclimation to the enclosed water masses and the Pacific.
**2.6 Sample processing, measurements, and data analyses**
All samples were further processed in a temporary laboratory in Club Náutico Del Centro Naval
and the Instituto del Mar del Perú (IMARPE). Sediment trap samples were processed directly
after the sampling boats returned to the harbor. First, the sample weight was determined
gravimetrically. Afterwards, the 5 L bottles were carefully rotated to re-suspend the material to
take homogenous subsamples from the collected particle suspensions for additional analyses
(e.g. particle sinking velocity) described in other papers of this special issue. The remaining
sample (always > 88 %) was enriched with 3 M $FeCl_3$ and 3 M NaOH (0.12 μl and 0.39 μl,
respectively per gram of sample) to adjust the pH to 8.1. The $FeCl_3$ addition initiated
flocculation and coagulation with subsequent sedimentation of particles within the 5 L bottle
(Boxhammer et al., 2016). After 1 hour, most of the supernatant above the settled sample was
carefully removed and remaining sample was centrifuged in two steps: 1) for 10 minutes at
~5200 g in a 800 mL beaker using a 6-16 KS centrifuge (Sigma);  2) for 10 minutes at ~5000 g
in a 110 mL beaker using a 3K12 centrifuge (Sigma). The supernatants were removed after both
steps and the remaining pellet was frozen at -20°C. The remaining water was removed by
freeze-drying the sample. The dry pellet was ground in a ball mill to generate a homogenous
powder which was fully recovered from the grinding chamber (Boxhammer et al., 2016).
Sub-samples of the powder to determine TPC and PON content were transferred into tin
capsules, weighed, and measured with an elemental analyzer following Sharp (1974). POC sub-
samples were treated identically but put into silver instead of tin capsules, acidified for 1 hour
with 1 M HCl to remove any particulate inorganic carbon, and dried at 50°C overnight. TPP
sub-samples were autoclaved for 30 minutes in 100 mL Schott Duran glass bottles using an
oxidizing decomposition solution (Merck, catalogue no. 112936) to convert organic P to
orthophosphate. P concentrations were determined spectrophotometrically thereafter following
Hansen and Koroleff (1999). BSi sub-samples were leached by alkaline pulping with 0.1 M





NaOH at 85°C in 60 mL Nalgene polypropylene bottles. After 135 minutes the leaching process
was terminated with 0.05 M $H_2SO_4$ and the dissolved Si concentration was measured
spectrophotometrically following Hansen and Koroleff (1999). POC, PON, TPP, and BSi
concentrations of the weighed sub-samples were scaled to represent the total sample weight so
that we ultimately determined the total element flux to the sediment traps.
Suspended TPC, POC, PON, TPP, BSi, and pigment concentrations sampled with the IWS in
the water columns were immediately transported to the laboratory and filtered either onto pre-
combusted (450°C, 6 hours) glass-fibre filters (GF/F, 0.7 μm nominal pore size, Whatman;
POC, PON, TPP, pigments) or cellulose acetate filters (0.65 μm pore size, Whatman; BSi)
applying gentle vacuum of 200 mbar. The filtration volumes were generally between 100 - 500
mL depending on the variable amount of particulate material present in the water columns.
Samples were stored either in pre-combusted (450°C, 6 hours) glass petri dishes (TPC, POC,
PON), in separate 100 mL Schott Duran glass bottles (TPP), 60 mL Nalgene polypropylene
bottles (BSi), or in cryo-vials (pigments). After filtrations, POC and PON filters were acidified
with 1 mL of 1 M HCl, dried overnight at 60°C, put into tin capsules, and stored in a desiccator
until analyses in Germany at GEOMAR following Sharp (1974). TPC samples were treated
identically, except for the acidification step, and they were dried in a separate oven to reassure
that they remain out of contact with any acid fume. TPP and BSi filters in the glass and
polypropylene bottles, respectively were stored at -20°C until enough samples had accumulated
for one measurement run. P and Si were extracted within the bottles and measured thereafter as
described for the sediment powder (see previous paragraph). TPP and BSi measurements of
suspended material were made in the laboratory in Peru so that no sample transport was
necessary.
Pigment samples were flash frozen in liquid nitrogen directly after filtration and stored at -
80°C. The frozen pigment samples were transported from Peru to Germany on dry ice within 3
days by World Courier. In Germany, samples were stored at -80°C until pigment extraction as
described by Paul et al. (2015). Concentrations of extracted pigments were measured by means
of reverse phase high performance liquid chromatography (HPLC, Barlow et al., 1997)
calibrated with commercial standards. The contribution of distinct phytoplankton taxa to the
total chl-a concentration was calculated with CHEMTAX which classifies phytoplankton taxa
based upon taxon-specific pigment ratios (Mackey et al., 1996). The dataset was binned into
two CHEMTAX runs: One for surface layer and one for the deeper layer (section 2.4) As input





pigment ratios we used the values for the Peruvian upwelling system determined by DiTullio
et al. (2005) as described by Meyer et al. (2017).
Mesozooplankton samples were analyzed until at least 50 individuals of the most abundant taxa
were counted (Ayón et al., in. prep.). As usual, zooplankton abundances were calculated
assuming 100% filtering efficiency of the net, although it is well-known that variation among
samples is often high (18–560% for copepods) due to plankton patchiness and species-specific
motilities (Wiebe and Holland, 1968).
Samples for inorganic nutrients were filtered (0.45 µm filter, Sterivex, Merck) immediately
after they had arrived in the laboratories at IMARPE. The subsequent analysis was carried out
using an autosampler (XY2 autosampler, SEAL Analytical) and a continuous flow analyzer
(QuAAtro AutoAnalyzer, SEAL Analytical) connected to a fluorescence detector (FP-2020,
JASCO). $PO_4^{3-}$ and $Si(OH)_4$ were analyzed colorimetrically following the procedures by
Murphy and Riley (1962) and Mullin and Riley (1955), respectively. $NO_3^-$ and $NO_2^-$ were
quantified through the formation of a pink azo dye as established by Morris and Riley (1963).
All colorimetric methods were corrected with the refractive index method developed by
Coverly et al. (2012). Ammonium concentrations were determined fluorometrically (Kérouel
and Aminot, 1997). The limit of detection (LOD) was calculated from blank measurements as
blank + 3 times the standard deviation of the blank (Thompson and Wood, 1995) over the course
of the experiment (LOD $NH_4^+$ = 0.063 µmol $L^{-1}$, $NO_2^-$ = 0.054 µmol $L^{-1}$, $NO_3^-$ = 0.123 µmolL$^-$
$^1$, $PO_4^{3-}$ = 0.033 µmol $L^{-1}$, $Si(OH)_4$ = 0.336 µmol $L^{-1}$). The precision of the measurements was
estimated from the average standard deviation between replicates over the course of the
experiment ($NH_4^+$ = 0.027 µmol $L^{-1}$, $NO_2^-$ = 0.014 µmol $L^{-1}$, $NO_3^-$ = 0.033 µmol $L^{-1}$, $PO_4^{3-}$ =
0.016 µmol $L^{-1}$, $Si(OH)_4$ = 0.016 µmol $L^{-1}$). The accuracy was monitored by including certified
reference material (CRM; Lot-BW, Kanso) during measurements. The accuracy was mostly
within CRM ±5 %, and ±10 % in the worst case.
After transportation to the laboratory, TDN and TDP samples were gently filtered through pre-
combusted (5 h, 450°C) Whatman GF/F filters (pore size 0.7 µm) using a diaphragm metering
pump (KNF Stepdos, continuous flow of 100 mL min$^{-1}$). The filtrate was collected in 50 mL
acid-cleaned HDPE bottles and immediately frozen at -20°C until further analysis. For the
determination of organic nutrient concentrations, filtered samples were thawed at room
temperature over a period of 24 hours and divided in half. One half was used to determine
inorganic nutrient concentrations as described above. The other half was used to determine



TDN and TDP concentrations. In order to liberate inorganic and oxidise nutrients, an oxidizing
reagent (Oxisolv, Merck) was added to samples, and these were subsequently autoclaved for 30
minutes and analyzed spectrophotometrically (QuAAtro, Seal Analytical). DON concentrations
were calculated by subtracting inorganic nitrogen ($NO_3^-$ and $NO_2^-$) from total dissolved
nitrogen (TDN). DOP was calculated as the difference between TDP and $PO_4^{3-}$.
Water samples for trace metal analysis were syringe filtered (0.20 µm, Millipore) into 125 mL
low density polyethylene (LDPE) bottles which were precleaned sequentially with detergent (1
week), 1.2 M HCl (1 week) and 1.2 M $HNO_3$ (1 week) with deionized water rinses between
each stage, and then stored in LDPE bags until required. Syringes/filters were precleaned with
0.1 M HCl. Samples were acidified with 180 µL HCl (UPA, Romil) in a laminar flow hood
upon return to the laboratory and allowed to stand >12 months prior to analysis. Dissolved trace
metal concentrations were determined following offline preconcentration on a Seafast system
via inductively coupled plasma mass spectrometry, exactly as per Rapp et al. (2017).
**3   Results**
**3.1 Physicochemical conditions in the water columns**
The water columns enclosed at the beginning of the study were temperature stratified with a
thermocline roughly at 5 m (Fig. 3). Surface temperatures were unusually high (up to 25°C)
during most of the first 40 days due to a rare coastal El Niño event which took place in austral
summer 2017 (Garreaud, 2018) (the last one prior to this was recorded in 1925 (Takahashi and
Martínez, 2017)). The coastal El Niño event ceased towards the end of the experiment (i.e.
beginning of April, ~day 38) and surface temperatures went back to more typical values for this
time of the year (<20°C). When averaged over the entire water column in all mesocosms,
temperatures ranged between 18.4 and 20.2°C from days 1 to 38 and between 17.9 and 18.6°C
thereafter. Temperature profiles were very similar in- and outside the mesocosms due to rapid
heat exchange (Fig. 3).
The salinity in the mesocosms was initially between 35.16 – 35.19, with little variation over the
19 m water column (Fig. 3). NaCl brine additions to below 10 m on days 13 and 33 (section
2.3) increased the salinity in the bottom layer (~10 – 17 m) to ~36.1 and ~36.4, respectively.
The salinity stratification stabilized the water column but sampling operations during the
experiment gradually mixed bottom water into the surface layer so that the salinity also
increased above 10 m. When averaged over the entire water column, salinities were between





35.16 – 35.24 until day 13, 35.57 – 35.67 between days 13 and 33, and 35.84 – 35.95 thereafter.
The salinity in the water outside the mesocosms was relatively stable around an average of
35.17 with 3 fresher periods in the surface layer due to river water inflow (Fig. 3).
The highest photon flux density measured at the surface inside the mesocosms (~0.1 m depth)
around noon time were ~500 – 600 $\mu mol\ m^{-2}\ s^{-1}$. PAR was on average about 35 % lower inside
the mesocosms than outside due to shading by the flotation frame and the bag. Figure 3 shows
light profiles relative to surface values (instead of absolute values) because CTD casts were
conducted at slightly different times of day and would therefore not be comparable on an
absolute scale. Light attenuation with depth was pronounced due to the high particle
concentrations in the water. Inside the mesocosms, 10 and 1% incident light levels were
generally shallower than 5 and 10 m. Outside, they were at slightly greater depths (Fig. 3).
Dissolved $O_2$ concentrations ($dO_2$) in- and outside the mesocosms were decreasing from >200
$\mu mol\ L^{-1}$ at the surface to <50 $\mu mol\ L^{-1}$ at depth (Fig. 3). The oxycline inside the mesocosms
was between 5 and 15 m. Oxycline depths were more variable outside the mesocosms where
low $dO_2$ events occurred more frequently in the upper water column. OMZ waters collected
from nearby stations 1 and 3 (Fig. 1) were added to the mesocosms on days 11 and 12. The
water column mixing as a consequence of the OMZ water addition led to the decrease of $dO_2$
in the surface layer and an increase of $dO_2$ in the lower water columns of the mesocosms. After
day 12, the salinity stratification stabilized the vertical $dO_2$ gradient which remained relatively
constant until the end of the experiment. Optode measurements had an offset of +13 $\mu mol\ L^{-1}$
in the bottom layer (15 m) and -16 $\mu mol\ L^{-1}$ in the surface (1 m) relative to the Winkler
measurements. Thus, there are inaccuracies of ±10-20 $\mu mol\ L^{-1}$. These inaccuracies were most
likely due to limitations associated with the response time of the sensor and therefore non-
random but led to carry-over along gradients. Nevertheless, the general trend observed in the
vertical $dO_2$ gradient as well as changes over time should be correctly represented in the present
dataset.

### 3.2 Inorganic and organic nutrients

$NO_3^- + NO_2^-$ concentrations ($NO_x^-$) in the mesocosms were initially between 5.6 – 7.6 $\mu mol\ L^{-1}$
and decreased in all mesocosms to 1.1 – 5.5 $\mu mol\ L^{-1}$ on days 11 and 12 (Fig. 4A). After the
OMZ water addition, $NO_x^-$ increased slightly in M2, M3, M6, and M7 (Fig. 4A, blue symbols)
as the OMZ source water from station 3 contained 4 $\mu mol\ L^{-1}$ of $NO_x^-$. M1, M4, M5, and M8
received OMZ water from station 1 with 0.3 $\mu mol\ L^{-1}$ and therefore $NO_x^-$ decreased in the days





following the OMZ water addition and reached the detection limit (i.e. 0.2 µmol L$^{-1}$ for NO$_3^-$)
between days 18 (M7) and 36 (M4). NO$_x^-$ was between 2.7 – 19.2 µmol L$^{-1}$ in the Pacific at the
deployment site and particularly high during the second half of the experiment (Fig. 4A).
PO$_4^{3-}$ concentrations in the mesocosms were initially between 1.4 – 2 µmol L$^{-1}$ and converged
to ~1.6 µmol L$^{-1}$ in all mesocosms 5 days after the start of the experiment (Fig. 4B). The OMZ
water contained 2.5 µmol L$^{-1}$ of PO$_4^{3-}$ at both stations so that its addition increased the PO$_4^{3-}$
concentrations in the mesocosms to 2 µmol L$^{-1}$. Afterwards, PO$_4^{3-}$ decreased in all mesocosms
but generally more profoundly in M2, M3, M6, and M7 (blue symbols in the figures) where
slightly more NO$_x^-$ was added through the OMZ water addition. PO$_4^{3-}$ decreased during the
second half of the experiment and was between 1.3 – 1.8 µmol L$^{-1}$ at the end. PO$_4^{3-}$ was between
1.5 – 3.1 µmol L$^{-1}$ in the Pacific and generally higher than in the mesocosms (Fig. 4B).
Si(OH)$_4$ concentrations in the mesocosms were initially between 6.1 – 10.3 µmol L$^{-1}$ and
decreased in all mesocosms until day 6 to values between 4.5 – 5.1 µmol L$^{-1}$ (Fig. 4C). The
OMZ water at station 1 and 3 contained 17.4 and 19.6 µmol L$^{-1}$ of Si(OH)$_4$, respectively, so
their additions increased the concentrations to 7.5 – 9.5 µmol L$^{-1}$ inside the mesocosms.
Concentrations remained quite stable at this level until day 26, after which they decreased in all
mesocosms to 2.5 – 4.5 µmol L$^{-1}$ at the end of the study. Si(OH)$_4$ was between 6.6 – 18.7 µmol
L$^{-1}$ in the Pacific and generally higher than inside the mesocosms, except for a few days (Fig.
4C).
NH$_4^+$ concentrations were initially between 2.2 – 5.5 µmol L$^{-1}$ and decreased to values <2 µmol
L$^{-1}$ on days 2 – 3 (Fig 4D). NH$_4^+$ increased thereafter (except for M8) to reach 1.5 – 2.4 µmol
L$^{-1}$ on day 10, but decreased again after the OMZ water additions to values close to or below
the limit of detection on day 18. Concentrations remained at a low level but increased slightly
by the end of the experiment to values between 0.1 – 1.4 µmol L$^{-1}$. NH$_4^+$ concentrations ranged
between the limit of detection and 7.1 µmol L$^{-1}$ in the Pacific and coincidently showed a similar
temporal pattern as in the mesocosms except for the time between days 10 and 20 where the
concentrations were considerably higher (Fig. 4D).
DON concentrations in the mesocosms were initially between 10.1 – 11.5 µmol L$^{-1}$ and
remained roughly within this range until the OMZ water addition. Afterwards, DON decreased
to 6 – 7.9 µmol L$^{-1}$ on day 30 but increased almost exponentially until the end of the experiment
(Fig. 4E). DON in the Pacific was within a similar range as in the mesocosms until the OMZ-





water addition, but shifted to a higher concentrations ($10 – 13.6$ µmol L$^{-1}$) from day 16 to 22,
followed by an abrupt decrease to $2.8 – 11.5$ from day 24 until the end of the experiment.
DOP concentrations in the mesocosms were initially between $0.45 – 0.63$ µmol L$^{-1}$ but declined
rapidly to $0.16 – 0.25$ µmol L$^{-1}$ on day 8. DOP increased after the OMZ-water addition to 0.22
$– 0.38$ µmol L$^{-1}$ and remained roughly at this level until day 40 after which it began to increase
to $0.56 – 0.7$ µmol L$^{-1}$ towards the end of the experiment. There were several day-to-day
fluctuations consistent among the mesocosms and we cannot fully exclude that these are due to
measurement inaccuracies (Fig. 4F). DOP in the Pacific was initially similar to the mesocosms
but decreased even more pronounced in the first week of the study to reach undetectable levels
on day 8. It increased, as in the mesocosms, on day 13 and remained at $0.29 – 0.45$ µmol L$^{-1}$
until day 32. After a short peak of 0.77 µmol L$^{-1}$ on day 34, DOP declined to $0.08 – 0.28$ µmol
L$^{-1}$ until the end of the experiment.
DIN:DIP (i.e. $(NO_x^- + NH_4^+):PO_4^{3-}$) in the mesocosms was constantly below the Redfield ratio
(i.e. 16) and its development largely resembled that of $NO_x^-$ as the predominant nitrogen source
(compare Figs. 4A and G). It was initially $5.4 – 7.7$ and decreased to $0.04 – 0.37$ by day 26
where it remained until the end of the experiment. DIN:DIP in the Pacific was similar to the
mesocosms until day 13, but considerably higher ($2.2 – 11.2$) thereafter (Fig. 4G).
DON:DOP in the mesocosms was initially close to the Redfield ratio (i.e. 16) but increased to
$29.2 – 40.4$ until the OMZ-water addition. Afterwards, DON:DOP declined to values slightly
above the Redfield ratio and remained at this level until the end of the experiment. The
occasional fluctuations towards higher values reflect the fluctuations in DOP (compare Fig. 4F
and H). DON:DOP in the Pacific was mostly above the Redfield ratio and generally higher than
in the mesocosms. It was initially 21.1 and increased to 77.6 on day 6 followed by a rapid
decline back to initial values. Afterwards, DON:DOP increased from 21.1 to 61.8 on day 42
(with one exceptionally low value on day 30) but then decreased to 19.5 at the end of the
experiment (Fig. 4H).
Dissolved iron (Fe) concentrations were generally elevated across all mesocosms with
concentrations ranging from 3.1 to 17.8 nM (Supplementary Table 1). The resolution of trace
metal clean sampling was insufficient to discuss the temporal trends in detail, although surface
concentrations appeared to be lower on day 48 (3.1-9.5 nM) than on day 3 (range 5.7-10.8 nM).
Dissolved Fe concentrations in Pacific water on day 48 (8.5 nM) were within the range of the
mesocosms and also comparable to the nanomolar concentrations of dissolved Fe reported



elsewhere in coastal surveys at shallow stations on the Peruvian Shelf (Bruland et al., 2005;
Chever et al., 2015).

### 3.3 Phytoplankton and zooplankton development

Chl-a concentrations in the mesocosms were initially between 2.3 – 4.9 µg L$^{-1}$ and declined to
1.4 to 2.4 µg L$^{-1}$ on day 8 (Fig. 5A). Initially high values of chl-a were found mostly above 5
and below 15 m (Fig. 5B). The OMZ water addition increased chl-a to 3.7 – 5.6 µg L$^{-1}$
(mesocosm-specific averages between days 12 – 40) except for M3 where concentrations
increased with a 1-week delay (3.4 µg L$^{-1}$ between days 22 – 36) and M4 where concentrations
remained at 1.6 µg L$^{-1}$ (average between days 12 – 40) and were largely unaffected by the
OMZ-water addition (Fig. 5A). The chl-a maximum remained in the upper 5 m in the week
after the OMZ-water addition but shifted to the intermediate depth range between 5 – 15 m
thereafter and remained there until approximately day 40. (Please note that the "quenching
effect" influences chl-a values especially near the surface so that absolute values may be biased;
see section 4.2). The exception was M4 where no such pronounced maximum was observed at
intermediate depths (Fig. 5B). Chl-a increased in all mesocosms, except for M4, to values up
to 38 µg L$^{-1}$ in the time after day 40 to the end of the experiment. This bloom occurred in the
uppermost part of the water column, due to surface eutrophication by defecating sea birds (Inca
Tern, *Larosterna inca*), who discovered the mesocosms as a suitable resting place (see section
4.1). Chl-a in the Pacific was initially within the range enclosed inside the mesocosms and
concentrations increased to slightly higher values around the same time as in the mesocosms
(Fig. 5). Throughout the study, chl-a in the Pacific was between 1.2 – 10.6 µg L$^{-1}$ with the chl-
a maxima always above 10 m (Fig. 5B).
The phytoplankton community composition was determined based on pigment concentration
ratios using CHEMTAX. We distinguished between seven phytoplankton classes: Chloro-,
Dino-, Crypto-, Cyano-, Prymnesio-, Pelagophyceae and diatoms (Figs. 6, S1) and use the word
"dominant" in the following when a group contributes >50 % to chl-a. Diatoms initially
dominated the community and contributed 50 – 59 % to the total chl-a concentration but
declined after the start while Chlorophyceae (or Dinophyceae in M1 and M7) became more
important. The other groups contributed mostly <25 % to chl-a before the OMZ water addition.
Diatoms contributed marginally to the chl-a increase in the days after the addition. Instead,
Dinophyceae became dominant in most mesocosms and contributed between 64 – 76 % to the
total chl-a until the end of the experiment (range based on averages between days 12 – 50



excluding M3 and M4). Imaging flow cytometry and microscopy revealed that the
dinoflagellate responsible for this dominance was the large (~60 μm) mixotrophic species
*Akashiwo sanguineum* which was present in abundances between ~ 40 – 100 cells mL$^{-1}$ (data
not shown). M3 and M4 were exceptions to this as Cryptophyceae rather than Dinophyceae
became dominant in the 10 days after the addition (Fig. 6). In M3, Dinophyceae became about
as dominant as in the other mesocosms when Cryptophyceae disappeared while they never
proliferated in M4. Chlorophyceae were detectable in all mesocosms after the OMZ addition
with relatively low chl-a contribution except for M1, M3, and M4 where they contributed up to
21, 78, and 98 %, respectively. Cyano-, Prymnesio-, and Pelagophyceae made hardly any
contribution to chl-a after the OMZ addition (average <3 %) except for M4 where they were
slightly more important (average = 7 %). Diatoms formed blooms in some mesocosms after day
30 where they became more important for relatively short times (M2, M5, M7, M8). The
phytoplankton community composition in the Pacific differed from that in the mesocosms.
Here, diatoms were dominant throughout the study period except for two very short periods
where either Chloro- + Dinophyceae (day 30) or Cyano- + Cryptophyceae dominated (day 36;
Fig. 6).
The mesozooplankton (MesoZP) community comprised various taxonomic groups among
which copepods were the predominant one. We therefore focus our analysis on them but point
towards a specific zooplankton analysis with more taxonomic detail provided in the framework
of this special issue (Ayón et al., in. prep.). All copepod species were pooled in three
developmental stages: nauplii, copepodites, and adults. The three main genera were
*Paracalanus*, *Hemicyclops*, and *Oncaea*, which can be considered omnivorous in a very wide
sense (Ayón et al., in. prep.).
In general, it was difficult to reveal clear population developments in this pooled dataset due to
considerable day-to-day fluctuations in the measured abundances (Fig.7). These fluctuations
are often found in MesoZP datasets and can be due to difficulties associated with net sampling,
counting uncertainties, and the patchy distribution of MesoZP in the water column (Algueró-
Muñiz et al., 2017; Lischka et al., 2017). Nevertheless, we observed a few temporal trends that
were sufficiently clear (and consistent with other datasets) so that we are confident that they
were "real" and outside the noise of the measurement. Most strikingly, copepod nauplii were
extremely low during almost the entire experiment. Some higher nauplii abundances occurred
on day 30 in the Pacific as well as towards the end of the study in M4 and M3. This increase in
copepod offspring co-occurred with a deepening of hypoxic layers (< 55 μmol L$^{-1}$) from ~10 m



to 14 – 15 m. Similarly, a short intrusion of higher oxygen waters up to ~10 m occurred in the
Pacific concomitantly with the minor nauplii increase on day 30 (Fig. 7C). Aside from this, the
copepod community seemed to stagnate with respect to developmental succession.
**3.4 Particulate matter pools and export fluxes**
POC concentrations in the mesocosm water columns ($POC_{WC}$) were initially between 49 - 66
µmol $L^{-1}$ and declined following the OMZ-water addition to 32 – 54 µmol $L^{-1}$ on day 16. $POC_{WC}$
started to increase after day 16 and $POC_{WC}$ reached a new steady state of 75 – 116 µmol $L^{-1}$
between days 24 and 44. Exceptions were M3 and M4 where the increase was either delayed
(M3) or did not take place at all (M4). $POC_{WC}$ increased rapidly at the end of the experiments
(Fig. 8A). $POC_{WC}$ in the Pacific was between 34 – 72 µmol $L^{-1}$ between days 0 – 24 and
decreased thereafter to values between 27 – 55 µmol $L^{-1}$ (Fig. 8A). The accumulation of POC
in the sediment traps ($\Sigma POC_{ST}$) was surprisingly constant over the course of the study, with an
average rate of 1.06 µmol POC $L^{-1}$ $d^{-1}$ (Fig. 8C).
$PON_{WC}$ concentrations in the mesocosms were initially between 9.2 – 11.9 µmol $L^{-1}$ and
declined after the OMZ-water addition to 6.2 – 10.3 µmol $L^{-1}$ on day 16. The increase in $PON_{WC}$
to 8.4 – 18.1 µmol $L^{-1}$ during days 17 – 24 was much less pronounced compared to $POC_{WC}$
(compare Figs. 9A and B). Furthermore, there was not such a pronounced difference to M3 and
M4 during this period where the development was similar as in the other mesocosms. However,
M4 was the only mesocosm where $PON_{WC}$ declined profoundly after day 30 and it remained at
a lower level until the end. $PON_{WC}$ in all other mesocosms remained at 5 – 18.1 µmol $L^{-1}$
between days 24 – 42 but increased markedly towards the end of the experiment (Fig. 8B).
$PON_{WC}$ in the Pacific varied between 7.9 – 16.2 µmol $L^{-1}$ between days 0 – 30 and 4.8 – 9.6
µmol $L^{-1}$ from day 32 until the end of the experiment. $\Sigma PON_{ST}$ accumulation was, like $\Sigma POC_{ST}$,
relatively constant over time, averaging at a rate of 0.15 µmol PON $L^{-1}$ $d^{-1}$ (Fig. 8D).
$BSi_{WC}$ concentrations in the mesocosms were initially 2.5 – 3.7 µmol $L^{-1}$ but decreased after
the OMZ-water addition to 0.4 – 0.8 µmol $L^{-1}$ on day 26. They remained at these low levels
until the end of the experiment with smaller peaks in some mesocosms due to minor diatom
blooms (compare Figs. 9D and 6). The $BSi_{WC}$ development in the Pacific was very different
from that in the mesocosms. Here, $BSi_{WC}$ was initially lower but increased to 6.4 between days
0 – 18. Afterwards it decreased for a short period but increased again towards the end of the
experiment (Fig. 8C). The $BSi_{ST}$ accumulation rate in the sediment traps was high in the first 3



weeks when diatoms were still relatively abundant (0.22 µmol BSi L$^{-1}$ d$^{-1}$), but very low
thereafter (0.04 µmol BSi L$^{-1}$ d$^{-1}$) (Fig. 8G).
TPP$_{WC}$ concentration decreased from 0.49 – 0.67 on day 0 to 0.27 – 0.36 µmol L$^{-1}$ on day 12
and remained around this level until day 20. Afterwards, TPP$_{WC}$ increased rapidly in all
mesocosms except M4 to a new level between 0.37 – 0.65 µmol L$^{-1}$ until day 24. TPP$_{WC}$
increased almost exponentially in all mesocosms from day 38 until the end of the experiment.
TPP$_{WC}$ was variable in the Pacific but generally higher between days 0 – 30 (0.37 – 0.77 µmol
L$^{-1}$) than from day 32 until the end (0.28 – 0.43 µmol L$^{-1}$) (Fig. 8D). ΣTPP$_{ST}$ accumulation was
constant at a rate of about 0.015 µmol TPP L$^{-1}$ d$^{-1}$ until day 40 but increased sharply to 0.1 µmol
TPP L$^{-1}$ d$^{-1}$ thereafter (Fig. 8H).
**3.5 Particulate organic matter stoichiometry**
POC$_{WC}$:PON$_{WC}$ in the mesocosms was initially between 5.1 – 5.8 and thus below the Redfield
ratio of 6.6. POC$_{WC}$:PON$_{WC}$ remained at approximately these values until some days after the
OMZ-water addition when it increased to 7.9 – 11.8 in all mesocosms except for M3 and M4.
In M3, the increase was delayed by about a week whereas it remained at a lower level of 3.5 –
8.3 in M4 throughout the experiment. POC$_{WC}$:PON$_{WC}$ decreased during the last ten days of the
study in all mesocosms except for M4 (Fig. 9A). POC$_{WC}$:PON$_{WC}$ in the Pacific remained around
the initial value of 6 throughout the study (Fig. 9A). POC$_{ST}$:PON$_{ST}$ ratios were considerably
less variable than POC$_{WC}$:PON$_{WC}$. They were initially 7.9 - 9 and therefore higher than in the
water column but decreased steadily over the course of the experiment so that they became
lower than in the water columns of most mesocosms (all except for M4) from around day 30
onwards (Fig. 9E).
POC$_{WC}$:TPP$_{WC}$ in the mesocosms was initially close to the Redfield ratio (i.e. 106) but increased
quite steadily up to 182 – 304 until day 38 except for a short decline after the OMZ-water
addition. The increase was also apparent in M3 and M4 even though it was less pronounced in
these two mesocosms and there was little change in the two weeks after the OMZ-water
addition. POC$_{WC}$:TPP$_{WC}$ decreased from days 40 to 44 when it reached values between 125 –
177 and remained approximately there (Fig. 9B). POC$_{WC}$:TPP$_{WC}$ was much more stable in the
Pacific and relatively close to the Redfield ratio throughout the experiment (Fig. 9B).
POC$_{ST}$:TPP$_{ST}$ was always considerably lower than POC$_{WC}$: TPP$_{WC}$ (compare Figs. 9B and F).
POC$_{ST}$:TPP$_{ST}$ increased in all mesocosms from initially 46 – 59 to 88 – 117 on day 18 after



which it varied widely between mesocosms. $POC_{ST}:TPP_{ST}$ converged to a much narrower and
very low value between 7 – 42 from day 40 until the end (Fig. 9F).
$POC_{WC}:BSi_{WC}$ in the mesocosms were between 8 – 34 from the start until day 16 but increased
substantially to 88 – 418 until day 28 and remained at a high level until the end of the
experiment. The increase in $POC_{WC}:BSi_{WC}$ was slightly delayed in M3 and generally less
pronounced in M4 (Fig. 9C). $POC_{WC}:BSi_{WC}$ in the Pacific remained at a low level of 7 – 38
throughout the experiment (Fig. 9C). $POC_{ST}:BSi_{ST}$ also increased around day 16 from 4 – 7
(until day 16) to 4 – 86 (day 18 until end) but was generally much lower than in the water
column throughout the study (compare Figs. 9C and G).
$PON_{WC}:TPP_{WC}$ in the mescosms was initially close to the Redfield ratio (i.e. 16) but increased
until the OMZ-water addition to 19 – 36. Afterwards $PON_{WC}:TPP_{WC}$ fluctuated around this
elevated value range with a slight tendency to decrease until the end of the experiment (Fig.
9D). $PON_{WC}:TPP_{WC}$ in the Pacific was 15 – 20 and thus mostly above the Redfield ratio until
day 24 but the positive offset increased to 15 – 32 thereafter (Fig. 9D). $PON_{ST}:TPP_{ST}$ was
considerably lower than $PON_{WC}:TPP_{WC}$ and below the Redfield ratio almost throughout the
experiment. Its temporal development largely resembled the development of $POC_{ST}:TPP_{ST}$
(compare Figs. 9F and H). It increased steadily from 6 – 7 at the beginning to 12 – 15 on day
18, followed by a phase of large variability between mesocosms until day 40. $PON_{ST}:TPP_{ST}$
converged to 1 – 5 afterwards and remained at this low level until the end of the experiment
(Fig. 9H).
**4    Discussion**
**4.1 Small scale variability, OMZ water signature similarities, and defecating seabirds:**
**Lessons learned from a challenging *in situ* mesocosm study during coastal El Niño 2017**
A key prerequisite to compare different mesocosm treatments is the enclosure of identical water
masses in all mesocosms at the beginning of the study (Spilling et al., 2019). Unfortunately,
this was not successful as can be seen for example in the initial inorganic nutrient concentrations
(Fig. 4). Although our procedure of lowering the mesocosms bags and allowing for several days
of water exchange does not exclude heterogeneity entirely (Bach et al., 2016; Paul et al., 2015;
Schulz et al., 2017), it was not as pronounced during our previous studies as experienced in
Peru. The reasons for this were likely the inherent small-scale patchiness of physicochemical
conditions which is a known feature in the near coastal parts of EBUS (Chavez and Messié,





2009). We encountered small foamy patches with H$_2$S smell indicative of sub-mesoscale
upwelling of anoxic waters, ultra-dense meter-sized swarms of zooplankton coloring the water
red, and brownish filaments of discharging river water from nearby Rio Rimac which carried
large amounts of water due to flooding during the coastal El Niño (Garreaud, 2018). In such
extraordinarily variable conditions, it may therefore be advisable to monitor the study site
before deployment and enclose water masses inside mesocosms within a very short time
opportunistically when conditions are relatively homogeneous within the study site.
A major motivation for our experiment was to investigate how plankton communities in the
coastal upwelling system off Peru would respond to upwelling of OMZ-waters with different
N:P signatures (question 2 mentioned in the introduction). The rationale for this was that
projected spatial extensions of OMZs and intensification of their oxygen depletion in a future
ocean could enhance the N-deficit in the study region with strong implications for ecological
and biogeochemical processes in the affected regions (García-Reyes et al., 2015; Stramma et
al., 2010). Unfortunately, however, there was unusually little bioavailable inorganic N in both
OMZ water masses when we collected them on days 5 and 10 so that the differences in inorganic
N:P signatures between the two treatments were minor after we had injected them into the two
sets of four replicate mesocosms (Fig. 4G). As a consequence, there was little potential to detect
treatment differences, especially in light of the large differences in the starting condition that
induced considerable variance between replicates (see previous paragraph). Because of these
difficulties we decided to focus on the analyses of temporal developments of ecological and
biogeochemical processes rather than on detecting treatment differences.
Another complicating factor in Peru was the presence of Inca Terns (*Larosterna inca*) – an
abundant sea bird species in the study region that was able to start and land on the limited space
between the anti-bird spikes we had installed on the mesocosm roofs (see video by Boxhammer
et al., 2019). They occasionally rested on the mesocosms until day 36 but their presence
increased abruptly thereafter. Additional bird scarers that we installed on all mesocosms on day
37 were unfortunately not preventing this from happening. During the last two weeks of the
study, we often counted more than 10 individuals on the floatation frames and the upper opening
of the bags. We noticed that they defecated into the mesocosms as there were excrements on
the inner sides of the bags above surface.
To get a rough estimate of the nutrient inputs through "orni-eutrophication" in the mesocosms
we first assumed that the increase of TPP export after day 40 is sinking excrement-P (Fig. 8H).



This assumption is reasonable because $PO_4^{3-}$ was far from limiting and did not show any
noticeable change in concentration during this time (Fig. 4B). Correcting the TPP-export after
day 40 (0.1 µmol $L^{-1}$ $d^{-1}$) with the background value in the time before (0.015 µmol $L^{-1}$ $d^{-1}$)
yields 0.085 µmol $L^{-1}$ $d^{-1}$ of P inputs from Inca Terns. This converts to 1.15 µmol $L^{-1}$ $d^{-1}$ of N
inputs, assuming a 13.5:1 N:P stoichiometry as reported for South American seabird excrements
(Otero et al., 2018). This estimation is in reasonable agreement with the observed $PON_{WC}$ +
DON + $NH_4^+$ increase of 5.2 – 17 µmol $L^{-1}$ observed from days 40 to 50 (Figs. 4D, E, and 8B;
note that $PON_{ST}$ as well as $NO_x^-$ are considered to remain constant in this approximation; Fig.
4A and 8F). These N-inputs into the mesocosms are at least 5 orders of magnitude higher than
what seabirds typically add to the water column of the Pacific in this region (Otero et al., 2018).
Accordingly, the phytoplankton bloom that occurred in the upper 5 m after day 40 was fueled
by orni-eutrophication. While this certainly is an undesired experimental artefact, it had some
advantages to interpret the data as will be highlighted in section 4.3.1.
The coastal El Niño that climaxed during our experiment (Garreaud, 2018) is the last peculiarity
we want to highlight in this section. Coastal El Niños are rare events with similar phenology as
usual El Niños, but regionally restricted to the far-eastern Pacific. The last such event of similar
strength occurred in 1925 (Takahashi and Martínez, 2017). Surface water temperatures (upper
5 m) are mostly below 20°C in this region during non El Niño years (Graco et al., 2017), but
were 20 – 25°C for most of the time during our study (Fig. 3A). This may have influenced
metabolic processes of plankton and also enhanced stratification. Thus, it is possible that the
observations discussed in the following sections may not be entirely representative for the much
more common non El Niño conditions.

### 4.2 Plankton succession

A new patch of upwelled water typically stimulates diatom proliferation (stronger than other
phytoplankton groups) as they have highest net growth rates under nutrient replete conditions
in turbulent environments (Moore and Villareal, 1996; Raven and Waite, 2004). A
dinoflagellate-dominated community typically follows when upwelling relaxes as they are
better adapted under more stratified conditions when motility and alternative nutrient
acquisition strategies such as mixotrophy play out as advantages (Smayda and Trainer, 2010).
This succession pattern was also observed in the mesocosms (except M3 and M4; see below),
where the initially enclosed nutrient-rich patch of water was occupied by diatoms followed by
the dinoflagellate *Akashiwo sanguinea* – a migratory and mixotrophic "harmful algal bloom"





(HAB)-forming species that is frequently observed in coastal environments including EBUS
(Badylak et al., 2014; Du et al., 2011; Jeong et al., 2005; Kudela et al., 2010; Park et al., 2002;
Smayda, 2010). The mesocosm environment with its reduced turbulence and enhanced
stratification through the brine addition to the bottom layer may have further promoted the *A.*
*sanguinea* blooms.
The addition of OMZ-water on days 11 and 12 had no obvious influence on the general
succession pattern because very little N, the limiting nutrient, was added. However, it likely
introduced new species into the mesocosms. Interestingly, short silicoflagellate blooms
occurred in some mesocosms after the OMZ-water addition, which we suspect to be important
for the BSi increase during this time (Grasse et al., in. prep.). The quasi absence of
silicoflagellates in M1 and M4 may have been related to trophic interactions as there were
pronounced copepod abundance peaks in M1 and M4 shortly before the silicoflagellate blooms
occurred. For a more detailed analysis of the role of silicoflagellates and their biogeochemical
foot print in this experiment please refer to Grasse et al., (in. prep.).
Exceptions to the succession pattern described above were M3 and especially M4. In M3, *A.*
*sanguinea* rose to dominance with a one week delay relative to the other mesocosms whereas
it never bloomed in M4. We assume that these differences were due to differences in the seeding
population of *A. sanguinea* which may have been lower in M3 and below the critical threshold
in M4 but we do not have data supporting this speculation. The comparison of mescocosms M3
and M4 with the others reveals the profound influence of *A. sanguinea* on the plankton food
web structure. Cryptophyceae were contributing considerably more to the bulk chl-a and were
able to form larger blooms in M3 and M4 when *A. sanguinea* was absent (Figs. 6, S1).
Furthermore, picoautotrophic (0.2 – 2 µm) Cyanophyceae and Chlorophyceae were able to form
major blooms in M4 (Figs. 6, S1). The absence of such blooms in the other mesocosms suggests
that they were controlled/suppressed by *A. sanguinea,* either through competition for resources
or grazing (Jeong et al., 2005).
Orni-eutrophication during the last 10 days of the experiment caused an unusual situation
because nutrients were not entering the euphotic zone through upwelling but were added to the
surface where light intensity was highest. The nutrients induced a considerable chl-a increase,
in the uppermost layer (Fig. S2) which was apparent in the chemical measurement (Fig. 5A).
The chl-a increase was less pronounced in the vertical profiles due to quenching effects in the
sunlit surface layer (Xing et al., 2012) and since the CTD probe misses the uppermost 0.3 m

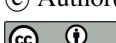



due to the sensor arrangement (Schulz and Riebesell, 2013). Therefore, chl-a surface
concentrations determined with the CTD probe must be interpreted with caution. The
CHEMTAX analysis implies that the chl-a increase was due to proliferating dinoflagellates
(Fig. S1). Flow cytometry and microscopy showed that it was not *A. sanguinea* but instead
some species in the nano-size class (i.e. $2 - 20$ µm; data not shown). The $2 - 4.5$ µmol L$^{-1}$
decrease in dissolved silic acid during this final period (day 36 until 50; Fig. 4C) implies that
diatoms were growing as well but this is at odds with the CHEMTAX data (Fig 6). It is also
inconsistent with BSi$_{WC}$ build up and BSi$_{ST}$ export during this period (day $36 - 50$), which
account for less than 25 % of the dissolved silicate drawdown (except for M4 where 85 % of
the drawdown is reflected in BSi$_{WC}$ buildup and BSi$_{ST}$ export). These inconsistencies in the Si
budget could be due to an internal storage of Si in diatoms that leaks out of the cells during
filtration and is therefore unaccounted in the budget as has been speculated by Boxhammer et
al. (2018). However, this remains speculative.
Copepods were the predominant mesozooplankton group throughout the experiment. They
were on average slightly more abundant in the mesocosms than in the Pacific. Additional
grazing assays of *Paracalanus* females, one of the dominant copepods, during the second half
of the experiment suggest that copepods guts were often empty and that they were not feeding
directly on phytoplankton as measured gut fluorescence was extremely low (Ayón et al., in.
prep.). These findings are supported by very low lipid contents of the copepods (*Paracalanus*,
*Hemicyclops*) with an almost absence of typical biomarker fatty acids, in particular diatom
markers (Ayón et al., in. prep.). This points to a community living at sub-optimal conditions.
The observed developmental delay of copepodites and adults and especially the very low
abundance of nauplii is presumably a consequence of hypoxic conditions in the mesocosms
below ~10 m (Fig. 3D). Despite species-specific tolerance levels, copepods generally respond
to hypoxic conditions with decreasing survival, egg production and population growth as well
as significant effects on population dynamics (Marcus et al., 2004; Richmond et al., 2006).
Sublethal and lethal hypoxia (< 67 and < 31 µmol L$^{-1}$ O$_2$, respectively,) occurred consistently
throughout the study in all mesocosms and Pacific below a depth around 10 m (Auel and
Verheye, 2007; Richmond et al., 2006). Particularly *Paracalanus* sp. may have been affected
by hypoxia as this species is a broadcast spawner releasing its eggs freely into the water column.
When sinking into hypoxic layers further development of eggs was likely impeded. Therefore,
slightly higher oxygen concentrations at the end of the study in M3 and M4 may have promoted
some egg/nauplii development in these mesocosms. However, these differences could also be



due to absence (M4) or lower prevalence (M3) of the HAB dinoflagellate *A. sanguinea* which
may have influenced nauplii development through trophic interactions.
The plankton food web in the Pacific was initially similar to the mesocosms but the diatom
dominance remained throughout the study period whereas the community changed profoundly
in the mesocosms (Figs. 6 and 7). The dissimilarity is not surprising as it is the consequence of
a fundamental difference in the sampling approaches. The mesocosms are geographically
stationary (Eulerian) but contain the same water mass for the entire experiment so that we were
sampling a Lagrangian model system. Thus, *in situ* mesocosms moored at a fixed position are
a hybrid between Eulerian and Lagrangian ("Eugrangian"). In contrast, the geographically
stationary sampling in the Pacific where the water masses flow along the sampling position is
Eulerian in the classical sense. Thus, in the Pacific we consistently monitored the early
succession stage dominated by diatoms, simply because the remaining succession occurred
further off shore. The Eularian sampling in the Pacific has therefore limited value to answer our
main question which is related to the plankton succession.
**4.3 Factors controlling productivity and export**
Messié and Chavez (2015) have identified light, macronutrient and iron supply as well as
physical processes (e.g. subduction) to be the key factors regulating primary and export
production in EBUS. We can immediately exclude physical processes and iron concentration
to have played a major role in our study. Physical processes above the micro-scale are excluded
in mesocosms. Iron concentrations are elevated to nanomolar concentrations in shallow waters
along the Peruvian shelf (Bruland et al., 2005) generally leading to a sharp contrast between
Fe-limited (or co-limited) offshore ecosystems and Fe-replete conditions in highly productive
inshore regions (Browning et al., 2018; Hutchins et al., 2002). Dissolved Fe concentrations
were verified to be high in the mesocosms both in surface and subsurface waters throughout the
experiment (days 3, 17, 48, Supplementary Table 1) confirming that Fe was replete compared
to N. Thus, our subsequent discussion will only consider light and macronutrients (mostly N
because P was also replete) as well as top-down control by grazing as controlling factors of
productivity and export.
**4.3.1. Productivity**
A remarkable observation is the decline in chl-a during the first 5 days despite high and
decreasing nutrient concentrations (Figs. 4 and 5). We explain this with the unusually high light



attenuation in the water column that was caused by a high standing stock of biomass in the
surface layer (Fig. 3C). Presumably, nutrients were quickly exhausted above the thermocline
(~5 – 10 m; Fig. 3A) where sufficient light allows fast growth so that further growth was
dependent on the limited nutrient supply that had to come from below. Conversely,
phytoplankton growth was restricted by light limitation below the mixed layer where nutrients
were likely more abundant. Thus, phytoplankton productivity was confined to a narrow depth
range mostly above the mixed layer so that loss processes (e.g. grazing and sedimentation),
when integrated over the entire water column, may have been dominant. Indeed, there is a
conspicuous chl-a peak in the funnels of the terminal sediment traps from days 3 to 10 which
points towards sinking of phytoplankton cells below the euphotic zone (Fig. 5B) – a loss process
that may have been amplified by the enclosure of the water column inside the mesocosms where
turbulence is reduced.
Chl-a was lowest during the OMZ water addition but increased in most mesocosms directly
afterwards due to the addition of inorganic N from the OMZ waters to the surface layer where
sufficient light was available. However, the OMZ water contained relatively little inorganic N
(~4 $\mu$mol L$^{-1}$ in the batch added to M2, M3, M6, M7 and ~0.3 $\mu$mol L$^{-1}$ in the batch added to
M1, M4, M5, M8) so that its potential to enhance productivity was limited. Interestingly, chl-a
did not noticeably increase in M3 and M4 (Fig. 5A) although inorganic N was consumed at
similar rates as in the other mesocosms (Fig. 4A and D). This difference could be due to a direct
channeling of autotrophic biomass into the microzooplankton pool or due to N uptake by
bacteria. Unfortunately, we have no data available to further explore these hypotheses.
*A. sanguinea* became dominant about one week after OMZ water addition when most inorganic
N sources were exhausted by species that grew in the previous week. This implies that *A.*
*sanguinea*, a facultative osmotroph (Kudela et al., 2010), extracted limiting N from the DON
pool, consistent with the decline in DON during days 15 - 25 (Fig 4E). The blooms of *A.*
*sanguinea* were associated with profound increase of $POC_{WC}$ and DOC of about 50 $\mu$mol L$^{-1}$,
respectively and a concomitant decrease of dissolved inorganic carbon (DIC) of ~100 $\mu$mol L$^{-}$
$^{1}$ (Fig. 8A; DOC data shown by Igarza et al., in. prep.; DIC data shown by Chen et al., in. prep.).
This is consistent with a considerable $dO_2$ increase above 100 % saturation in those mesocosms
harboring *A. sanguinea* (all except M4). Altogether, these data suggest that *A. sanguinea*
contributed significantly to organic carbon fixation in the mesocosms.



Another interesting observation with respect to *A. sanguinea* was its long persistence. It
consistently contributed the majority of chl-a after it had risen to dominance (Figs. 6, S1) and
even persisted during the orni-eutrophication event where other phytoplankton exploited the
surface eutrophication and generated additional POC (Fig. 8A). Importantly, *A. sanguinea*
contributed to a high level of chl-a even after the build-up of POC and DOC and the concomitant
draw-down of DIC, roughly between days 15 – 25, had stopped (Fig. 8A; DOC data shown by
Igarza et al., in. prep.; DIC data shown by Chen et al., in. prep.). This observation highlights
the difficulties when assessing productivity from chl-a (e.g. through remote sensing) because
mixotrophic species like *A. sanguinea* may conserve high pigment concentrations even when
photosynthetic rates are muted.
Orni-eutrophication during the last 10 days enabled rapid phytoplankton growth through the
relief from N-limitation and high light intensities in the uppermost meters. Grazers could not
control such rapid growth so phytoplankton generated an enormous chl-a peak even though
copepodites and adults increased in abundance in most mesocosms (Figs. 5A, 7A, B, and S2).
The fact that the bloom occurred so intensely in the surface highlights the role of light limitation
in the coastal Peruvian upwelling system. It appears that self-shading due to high biomass is a
key mechanism muting phytoplankton growth thereby enabling a close coupling between
productivity and loss processes as reflected in the relative constancy of chl-a, $POC_{WC}$ and
$POC_{ST}$ (Figs. 5A and 8A, E; see next section for further details on export). Indeed, when
limiting nutrients are added to a layer with high light intensity then phytoplankton can break
this coupling and realize rapid production, reflected in rapid chl-a upward excursions (Fig. 5A).
The Eulerian sampling of the Pacific did not allow us to observe succession patterns and the
build-up and decline of biomass because for this we would have needed to monitor the same
water mass over time. Thus, in order to compare and eventually assess the representativeness
of the mesocosm results for the wider region we would need "true" Lagrangian studies
following a patch of water from the location of upwelling to further offshore. In addition to
physical considerations, these Lagrangian studies would additionally have to consider the
effects of rapidly declining Fe concentrations with distance from the coastline on phytoplankton
succession. The mesocosm experiment herein was representative of highly productive inshore
waters where water upwelled over a broad shelf region contains very high concentrations of
~10 nM Fe. Yet Fe-deficient conditions are expected in regions where the shelf is narrower,
and generally moving further offshore.





### 4.3.2 Export flux


POC$_{ST}$ and PON$_{ST}$ export flux were remarkably constant over the course of the study (Fig. 8E,
F; the same applies for TPP$_{ST}$ export until day 40 after which bird defecation became
significant, Fig. 8H). As for productivity, we assume the constancy to be rooted in the N and
light co-limitation which mutes pulses of productivity and allows a closer coupling of
productivity with export. Mechanistically, this may be explained by a relatively constant
physical coagulation rate and/or a relatively constant grazer turnover establishing relatively
constant biologically mediated aggregation and sinking (Jackson, 1990; Wassmann, 1998).
Interestingly, M4 was not different to the other mesocosms even though the enormous POC$_{WC}$
build-up through *A. sanguinea* was absent (Fig. 8A, E). This observation implies a limited
influence of *A. sanguinea* on export production over the duration of the experiment.
Nevertheless, it is likely that the biomass generated by *A. sanguinea* would have enhanced
export flux when their populations started to decline and sink out. Unfortunately, we could not
observe the *A. sanguinea* sinking event as we had to terminate the study (day 50) before the
population declined. However, these findings allow us to conclude that the time lag between
the *A. sanguinea* biomass build-up (day ~15) and decay is at least 35 days. This is an important
observation as it implies that production by these types of dinoflagellates can be temporarily
and spatially highly uncoupled – a factor that is often neglected in studies of organic matter
export (Laws and Maiti, 2019; Stange et al., 2017).
Another interesting aspect with respect to the constancy of the POC$_{ST}$ and PON$_{ST}$ export flux
is the sharp decline of the BSi$_{ST}$ export flux around day 20 (Fig. 8G). This indicates that
sustaining a constant POC$_{ST}$ and PON$_{ST}$ export flux did not depend on diatoms. Furthermore,
cumulative $\Sigma$BSi$_{ST}$ and $\Sigma$POC$_{ST}$ on day 50 do not correlate across mesocosms, showing that
increased $\Sigma$BSi$_{ST}$ export does not necessarily enhance total $\Sigma$POC$_{ST}$ export (insignificant linear
regression; data not shown). Thus, silicifiers seem to have had a (perhaps surprisingly) small
influence on controlling POC$_{ST}$ export fluxes in the present experiment.

### 4.4 Particulate C:N:P:Si stoichiometry in the mesocosms


### 4.4.1 C:N


POC$_{WC}$:PON$_{WC}$ was mostly below the Redfield ratio (i.e. 6.6:1 mol:mol) until the OMZ water
addition (Fig. 9A). The low values coincide with the initial dominance of diatoms and these are
known to have an inherently lower particulate C:N stoichiometry than dinoflagellates (Quigg





et al., 2003). Yet, the absolute $POC_{WC}:PON_{WC}$ ratios are still at the lower end even for diatoms,
indicating that the predominant species had particularly low C:N and/or that growth conditions
(e.g. light limitation) led to a high N demand (Brzezinski, 1985; Terry et al., 1983).
$POC_{ST}:PON_{ST}$ was higher than $POC_{WC}:PON_{WC}$ during the initial period indicating preferential
remineralization of N over C. After the OMZ water addition, $POC_{WC}:PON_{WC}$ increased
substantially due to the *A. sanguinea* bloom. The predominant control of *A. sanguinea* on the
$POC_{WC}:PON_{WC}$ during this time is clear as we saw no increase in M4 where this species was
absent and a delayed increase in M3 where the *A. sanguinea* bloom was delayed. Importantly,
the increase of $POC_{WC}:PON_{WC}$ is not reflected in an increase of $POC_{ST}:PON_{ST}$ (Fig. 9 A, E).
This strongly supports our interpretations in section 4.3.2 that *A. sanguinea* did not notably
contribute to export production before the experiment was terminated because otherwise we
would have expected the $POC_{WC}:PON_{WC}$ signal to occur in the sediment traps as well. It also
suggests that the time lag between organic matter production and export is variable and depends
on the lifestyles of predominant primary producers (see section 4.3.2). During the last ten days,
both $POC_{WC}:PON_{WC}$ and $POC_{ST}:PON_{ST}$ declined despite the ongoing prevalence of *A.*
*sanguinea*. The decline was most likely triggered by the orni-eutrophication event which
fertilized a bloom with new nutrients in the uppermost water column (section 4.1).
**4.4.2 C:P**
$POC_{WC}:TPP_{WC}$ was initially close to the Redfield ratio (i.e. 106:1 mol:mol), but started to
increase in all mesocosms from early on until around day 40 (with a minor decrease after the
OMZ water addition, Fig. 9B). The increase was less pronounced but also present in M4 where
*A. sanguinea* did not bloom. This suggests that *A. sanguinea* was the main driver of this trend
but other players in the plankton communities responded similarly with respect to the direction
of change. Interestingly, there was a tendency of decreasing $POC_{WC}:TPP_{WC}$ during periods of
chl-a increase which may be due to the cells acquiring P for cell divisions (Klausmeier et al.,

940    2004).

$POC_{ST}:TPP_{ST}$ was considerably lower than $POC_{WC}:TPP_{WC}$ throughout the experiment
indicative for the unusual observation of preferential remineralization of C over P in the water
column. The extremely low $POC_{ST}:TPP_{ST}$ values recorded during the last 10 days of the
experiment are very likely due to the orni-eutrophication where defecated P sank unutilized into
the sediment traps.





### 946  4.4.3 C:Si

$POC_{WC}:BSi_{WC}$ was initially low (Fig. 9C), indicative for a diatom dominated community
(Brzezinski, 1985). The increase of $POC_{WC}:BSi_{WC}$ about a week after the OMZ-water addition
coincides roughly with the depletion of $NO_x^-$ even though $Si(OH)_4$ was still available in higher
concentrations (compare Figs. 4A, C and 9C). This suggests that the switch from a diatom to a
dinoflagellate predominance as seen in most mesocosms was triggered by N and not Si
limitation. The $POC_{WC}:BSi_{WC}$ increase is lower in M4 where *A. sanguinea* was absent,
underlining that this species was a key player driving the trend in the other mesocosms.
$POC_{ST}:BSi_{ST}$ was also increasing after the OMZ-water addition but considerably less
pronounced than $POC_{WC}:BSi_{WC}$. Once again, the explanation for this is the persistence of *A.*
*sanguinea* which maintains the high signal in the water column but does not transfer it to the
exported material because it did not sink out during the experiment.

### 958  4.4.4 N:P

$PON_{WC}:TPP_{WC}$ was higher than the Redfield ratio (i.e. 16:1) almost throughout the entire
experiment (Fig. 9D), although still within the range of what can be found in coastal regions
(Sterner et al., 2008) and among phytoplankton taxa (Quigg et al., 2003). The large positive
offset relative to the dissolved inorganic N:P ratio, which was initially 8:1 - 5:1 but then
decreased to values around 0.1:1, likely reflects that the plankton community has a certain N
requirement that is independent of the unusually high P availability. Hence, inorganic N:P may
not be a suitable predictor of particulate N:P under these extreme conditions.
Another interesting observation in this context was than $PON_{WC}:TPP_{WC}$ was increasing initially
even though the inorganic nutrient N:P supply ratio was decreasing (compare Fig. 4G and 9D).
This observation is inconsistent with a previous shipboard incubation study in the Peruvian
upwelling system (Franz et al., 2012b) and also contrary to our expectations based on meta-
analyses (Hillebrand et al., 2013). We can only speculate about the opposing trend between
inorganic N:P and $PON_{WC}:TPP_{WC}$ but consider stoichiometric changes bound to the
phytoplankton succession to be the most plausible explanation. Presumably, the transition from
diatoms with intrinsically low N:P towards Chlorophyceae and Dinophyceae with higher N:P
during the first ten days may largely explain this observation (Quigg et al., 2003).





Not surprisingly, $PON_{ST}:TPP_{ST}$ was lower that $PON_{WC}:TPP_{WC}$ indicating preferential
remineralization of the limiting N over the replete P in the water column. Additionally, the P
inputs from defecating birds during the last ten days mostly sank out unutilized and further
reduced the already low $PON_{ST}:TPP_{ST}$.

### 979   4.5 C:N:P:Si of suspended organic material in the Pacific

C:N:P:Si stoichiometry of suspended material was much more constant in the Pacific than in
the mesocosms (Fig. 9A – D). This observation is a consequence of the Eularian (i.e.
geographically stationary) sampling where regular upwelling and nutrient resupply conserved
the prevalence of a diatom-dominated early succession stage at the sampling location (see
section 4.2).
Perhaps the one noteworthy change was the observed increase in $PON_{WC}:TPP_{WC}$ after day 20
(Fig. 9D). This increase cannot be explained by a shift in community composition since diatoms
were dominant before and after day 20 (Fig. 6). However, we observed a pronounced increase
in the inorganic N:P nutrient ratio during this time, driven by an increase in N (Fig. 4A, G).
Thus, the $PON_{WC}:TPP_{WC}$ increase in the Pacific was consistent with the N:P supply ratio which
is in contrast to the mesocosms where $PON_{WC}:TPP_{WC}$ and inorganic N:P changed in an opposite
trend (section 4.4.4). We explain this inconsistent pattern with the fundamental differences in
the community development between the mesocosms and the Pacific. In the Pacific, diatoms
prevailed most of the time so that the higher inorganic N:P supply could have triggered a more
consistent physiological response towards higher $PON_{WC}:TPP_{WC}$. In the mesocosms, nutrient
resupply was cut off leading to major shifts in the community composition towards
dinoflagellates when the nutrients were exhausted (section 4.1). Thus, the shift in
$PON_{WC}:TPP_{WC}$ in the mesocosms was triggered by ecology whereas it was arguably triggered
by a physiological response in the Pacific.

### 999   5   Synthesis

This section synthesizes the most important patterns with respect to productivity, export, and
stoichiometry. Based on the processes described in the discussion we subdivide the mesocosm
experiment in 3 main phases (see Figure 10 for a synthesis graphic).
Phase 1 lasts from day 1 until the OMZ-water addition (days 10 and 12) and describes what we
would consider the expected early succession diatom dominated community. Here, diatoms



grow near the surface where they quickly exhaust inorganic N. Inorganic N is still available
deeper in the water column but low light availability limits growth rates so that loss processes
are higher than gains. Loss is likely due to grazing but also due to phytoplankton sedimentation
as indicated by a sharp chl-a peak in the sediment trap funnels below 17 m. The BSi export is
relatively high while the POC export is not, indicating that diatoms did not enhance organic
matter export compared to other communities prevailing later in the experiment. The C:N of
suspended matter is low whereas C:N of sinking material is higher, indicating high N demand
of the community (preferential remineralization of N). This is supported by the low (i.e. much
below the Redfield ratio) N:P.
Phase 2 lasts from the OMZ-water addition until day 40 and is characterized by the dominant
influence of the mixotrophic dinoflagellate *Akashiwo sanguinea*. It started rising to dominance
about one week after the OMZ-water addition, directly after a short bloom of silicoflagellates
and/or Cryptophyceae. The *A. sanguinea* bloom was fueled by inorganic and organic nutrients
and roughly doubled the amount of POC in the water column. However, the biomass formed
by this species did not sink out in significant quantities and remained in the water column until
the experiment was terminated. Thus, the export flux during the experiment was not different
in mesocosms where *A. sanguinea* bloomed compared to the one mesocosm (M4) where this
bloom did not occur, despite very large differences in productivity. These findings suggest that
productivity and export by mixotrophic dinoflagellates can be spatially and temporarily highly
uncoupled which is an important factor to consider when determining export ratios (i.e. export
production/primary production). Mesozooplankton could not capitalize on the new biomass
formed by *A. sanguinea*, possibly because *A. sanguinea* constituted an inappropriate food
source and/or low oxygen impeded mesozooplankton reproduction. The *A. sanguinea* bloom
also left a major imprint on particulate organic matter stoichiometry by increasing C:N, C:P,
and C:Si.
Phase 3 lasts from day 40 until the end of the experiment and is characterized by defecations of
the seabird *Larosterna inca* (Inca Tern) into the mesocosms. This orni-eutrophication triggered
intense phytoplankton blooms in most mesocosms in the uppermost part of the water column
where light was plentiful. N inputs through bird excrements were directly utilized and converted
into organic biomass whereas the defecated P remained unutilized and sank through the water
column directly into the sediment traps. *A. sanguinea* persisted during this bloom at
intermediate depth (~10 m) so the surface bloom added organic biomass to the already available
standing stock. Organic matter export (except for TPP) was not increasing during the bloom,





likely because the new biomass was still accumulating in the water column and the experiment
was terminated before it had the chance to sink out. The orni-eutrophication relaxed the N-
limitation, at least near the surface, so that suspended organic matter C:N and N:P decreased
and increased, respectively, relative to phase 2.
Sampling conditions in the Pacific were fundamentally different to the mesocosms because the
latter are stationary and contained the same water mass, whereas water at the sampling location
in the Pacific flows and is not stationary. Thus, plankton successions could be monitored in the
mesocosms but not in the Pacific because in the latter observations are confounded by changes
through advection. Therefore, plankton communities in the Pacific resembled an early, diatom-
dominated, succession stage since regular upwelling events provided nutrients continuously,
albeit at variable concentrations. A relaxation of upwelling and a transition to a later succession
stage would likely have been observed when the water traveled further offshore where
upwelling pulses become less and eventually cease.
Altogether, our study revealed some important factors controlling plankton productivity,
particulate matter stoichiometry, and export flux in the coastal upwelling system off Peru. These
findings will help to improve our mechanistic understanding of key processes in this region and
be valuable for modelling. The analysis provided in this paper covers many of the most
noticeable outcomes of this experiment with respect to ecology and biogeochemistry. However,
more specialized papers will be published within this Biogeosciences special issue and provide
additional detail on important aspects including: oceanographic conditions during the coastal
El Niño; phyto- and zooplankton succession patterns; microbial diversity; enzyme activities;
phytoplankton fatty acid profiles; archaeal lipidomes; carbonate chemistry; community
production and respiration; $N_2$ fixation; N loss processes; DOC dynamics; Si isotope
fractionation; Sinking velocity and export.
**Data availability**
All data will be made available on the permanent repository www.pangaea.de after publication.
**Author contribution**
LTB, AJP, TB, KGS, MH, AL, SL, CS, MS, UR designed the experiment. LTB, AJP, TB,
EvdE, KGS, Pag, IB, A-SB, S-HC, JC, KD, AF, MF, MH, JH, NH-H, VK, LK, PK, CL, SL,
JaM, JuM, FM, JP, CSf, KS, CSp, MS, MZM, UR contributed to the sampling. LTB, AJP, TB,



EvdE, KGS, EPA, JA, PAy, IB, AB, MH, VK, JL, SL, AL, JaM, JuM, FM, CS, SS analyzed
the data. LTB wrote the manuscript with comments from all co-authors.
**Competing interests**
The authors declare that they have no conflict of interests.
**Acknowledgements**
This project was supported by the Collaborative Research Centre SFB 754 Climate-
Biogeochemistry Interactions in the Tropical Ocean financed by the German Research
Foundation (DFG). Additional funding was provided by the EU project AQUACOSM and the
Leibniz Award 2012 granted to U.R. We thank all participants of the KOSMOS-Peru 2017
study for assisting in mesocosm sampling and maintenance. We are particularly thankful to the
staff of IMARPE for their support during the planning, preparation and execution of this study
and to the captains and crews of BAP MORALES, IMARPE VI and BIC HUMBOLDT for
support during deployment and recovery of the mesocosms and various operations during the
course of this investigation. Special thanks go to the Marina de Guerra del Perú, in particular
the submarine section of the Navy of Callao, and to the Dirección General de Capitanías y
Guardacostas. We also acknowledge strong support for sampling and mesocosm maintenance
by Jean-Pierre Bednar, Gabriela Chavez, Susanne Feiersinger, Peter Fritsche, Paul Stange,
Anna Schukat, Michael Krudewig. We want to thank Club Náutico Del Centro Naval for
excellent hosting of our temporary filtration laboratory, office space and their great support and
improvisation skills after two of our boats were lost. This work is a contribution in the
framework of the Cooperation agreement between the IMARPE and GEOMAR through the
German Ministry for Education and Research (BMBF) project ASLAEL 12-016 and the
national project Integrated Study of the Upwelling System off Peru developed by the Direction
of Oceanography and Climate Change of IMARPE, PPR 137 CONCYTEC.

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

**Figures and tables**

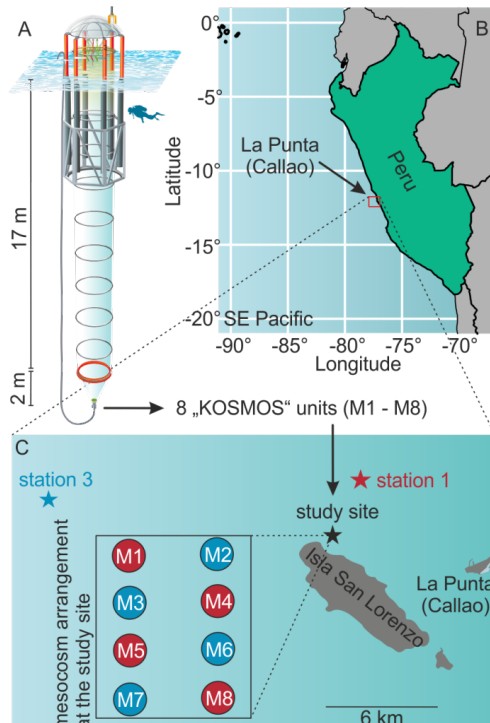

**Figure 1**. The mesocosm study site. (A) Graphic of one KOSMOS unit with underwater bag
dimensions given on the left. (B) Overview map of the study region. Please note that the square
marking the study site is not true to scale. (C) Detailed map of the study site. The laboratories
for sample processing were located in La Punta (Callao). Coordinates of relevant sites are given
in section 2.1.

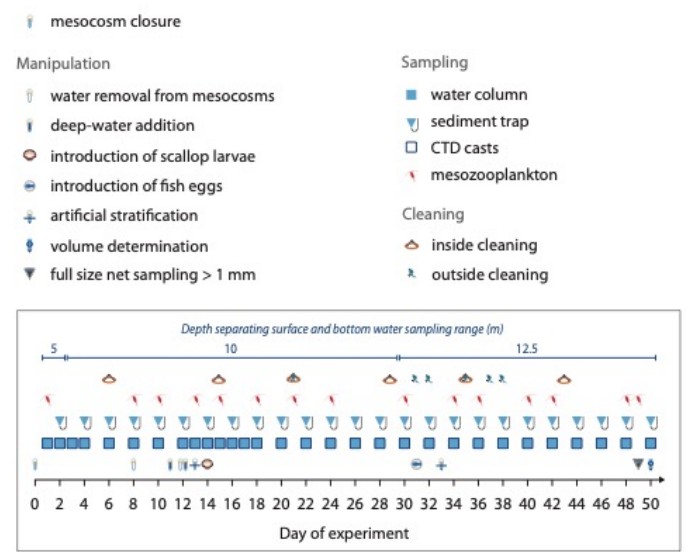


**Figure 2**. Manipulation, sampling, and maintenance schedule. Day 0 was February 25, 2017
and day 50 was April 16, 2017. Also given is the depth separating the surface and bottom waters
sampling range of the course of the study.




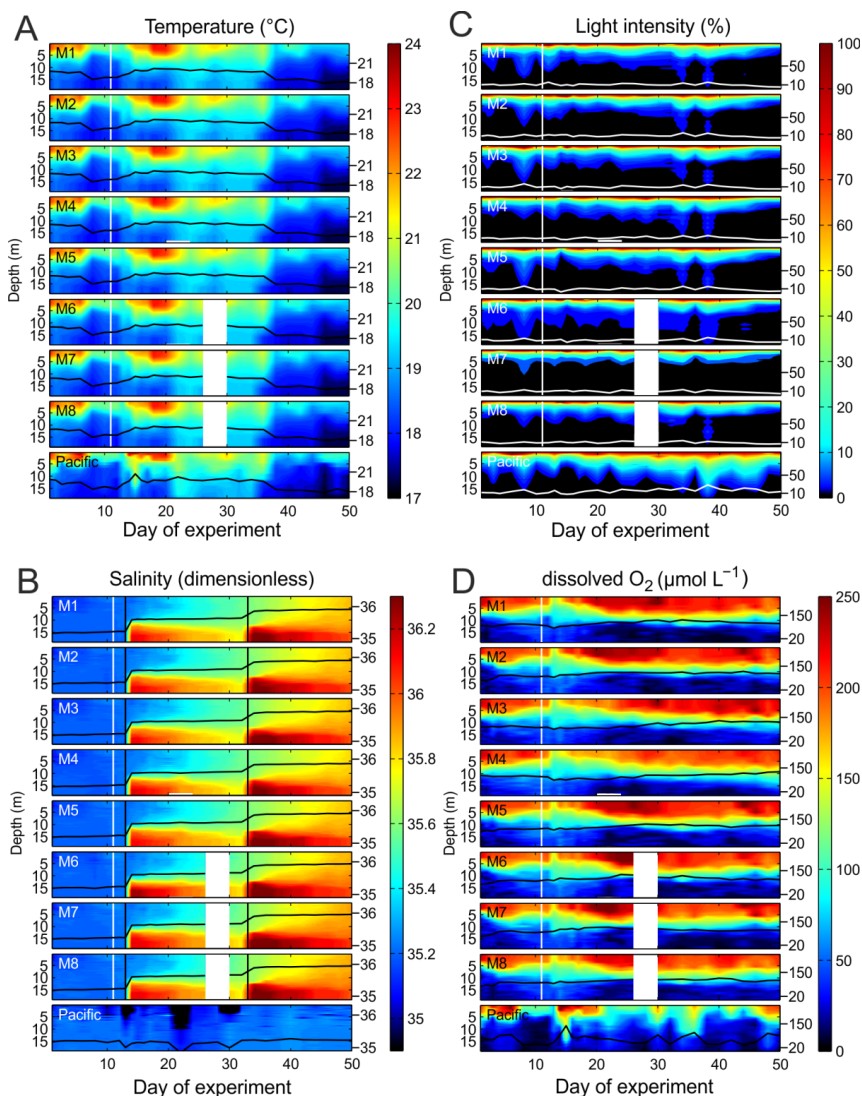

**Figure 3**. Physical and chemical conditions in the enclosed water columns of mesocosms M1 – M8 and the Pacific at the mesocosm mooring site determined with CTD casts. The black or white lines on top of the contours show the depth integrated water column average with the corresponding additional y-axes on the right side. The vertical white lines indicate the time of OMZ water additions to the mesocosms. The lack of data on day 28 in M6, M7, and M8 was due to problems with power supply. (A) Temperature in °C. (B) Salinity (dimensionless). The vertical black lines mark the NaCl brine additions. (C) Light intensity (photosynthetic active radiation) normalized to surface irradiance in the upper 0.3 m. (D) Dissolved $O_2$ concentrations.





1375

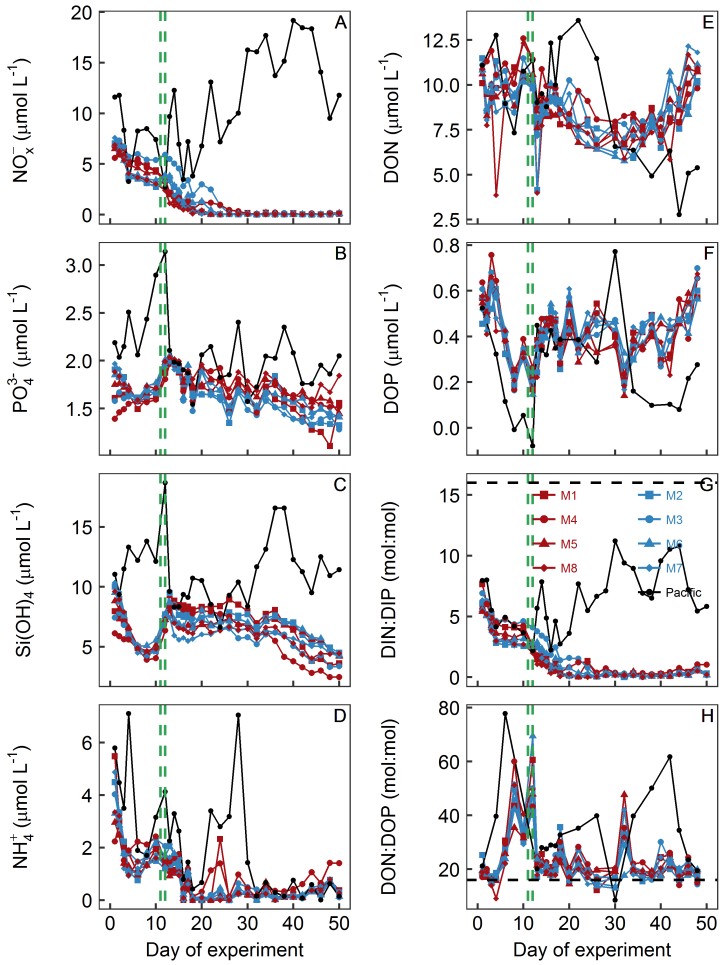

**Figure 4**. Inorganic and organic nutrient concentrations and stoichiometries integrated over the
0 – 17 m depth range. The horizontal dashed black line in panel (G) displays the Redfield ratio
of DIN:DIP = 16. The green lines mark the days of OMZ water additions.  (A) $NO_3^-$ + $NO_2^-$.
(B) $PO_4^{3-}$. (C) $Si(OH)_4$. (D) $NH_4^+$. (E) DON. (F) DOP. (G) DIN:DIP, i.e. $(NO_x^- + NH_4^+)/PO_4^{3-}$
. (H) DON/DOP.






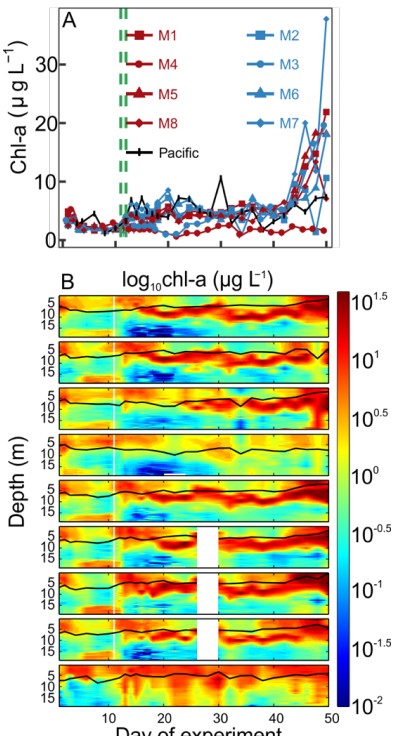


**Figure 5**. Chlorophyll a concentrations. (A) Average chl-a concentrations over the entire water
column (0 – 17 m) measured with HPLC. (B) Vertical distribution of chl-a determined with the
CTD fluorescence sensor. The offset of the CTD sensor was corrected with the HPLC chl-a
data. Please note, however, that the quenching effect may have influenced chl-a near the
surface.





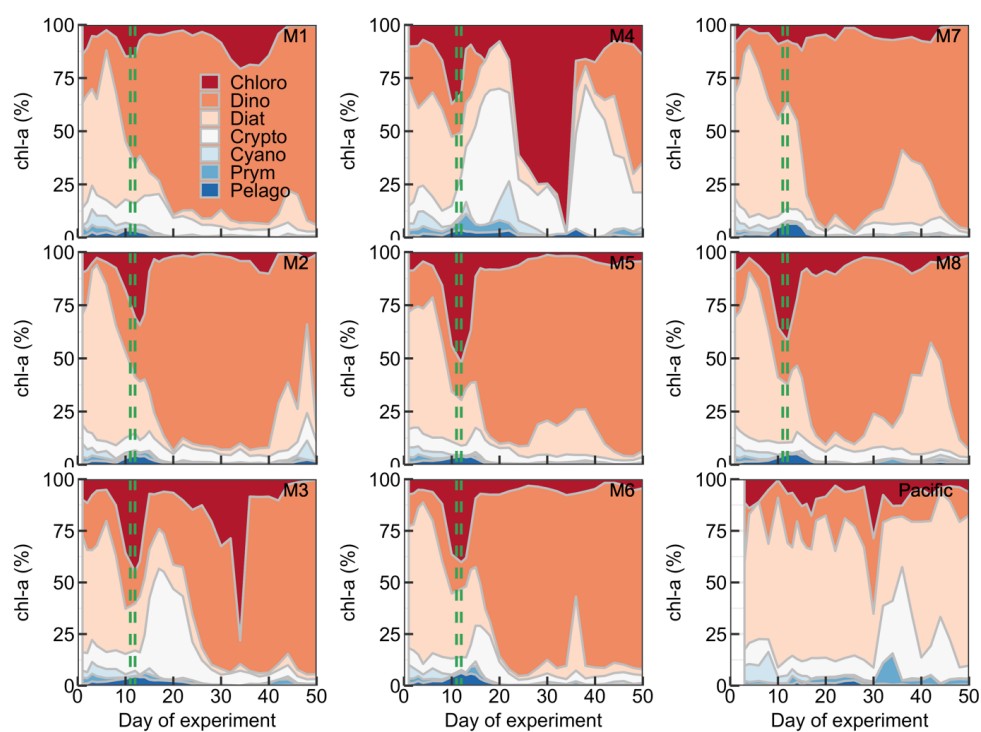

**Figure 6**. Relative contribution of the different phytoplankton classes to the total chl-a concentration. The mesocosm number is given on the top right of each subplot. The green lines mark the days of OMZ water additions.





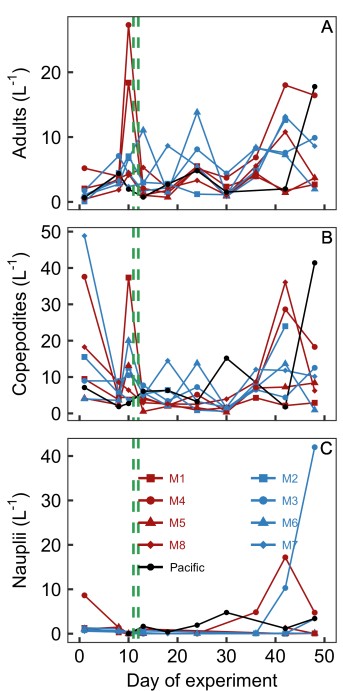


**Figure 7**. Copepod abundances. (A) Adults. (B) Copepodites. (C) Nauplii. Abundances shown
here are the sum of all species. By far the numerically dominant genera were *Paracalanus*
(mostly *Paracalanus parvus*) and *Hemicyclops*.
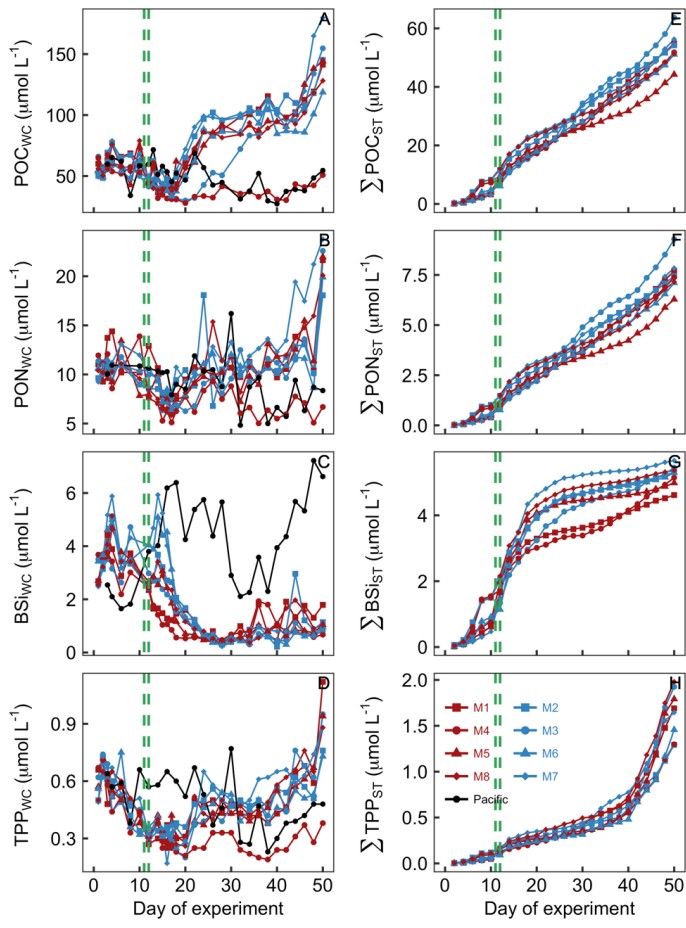


**Figure 8**. Particulate organic matter concentrations and cumulative export. Shown in the left column (A – D) are concentrations averaged over the entire water column (0 – 17 m). Shown in the right column (E – H) are cumulative export fluxes of particulate matter over the course of the study. The green lines mark the days of OMZ water additions.




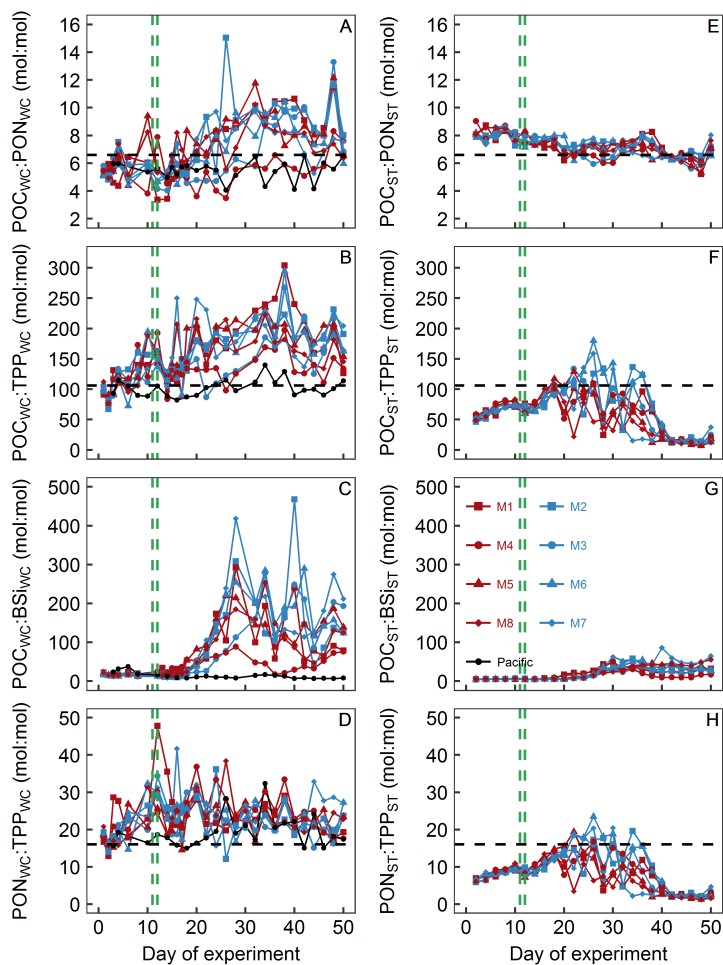


**Figure 9**. Particulate matter stoichiometry. Shown in the left column (A – D) are elemental ratios of particulate matter in the water column. The right column (E – H) shows the same ratios but for particulate matter collected in the sediment traps. The horizontal dashed black lines display Redfield ratios (i.e. POC:PON = 6.6, POC:TPP = 106, PON:TPP = 16). The vertical dashed green lines mark the days of OMZ water additions.


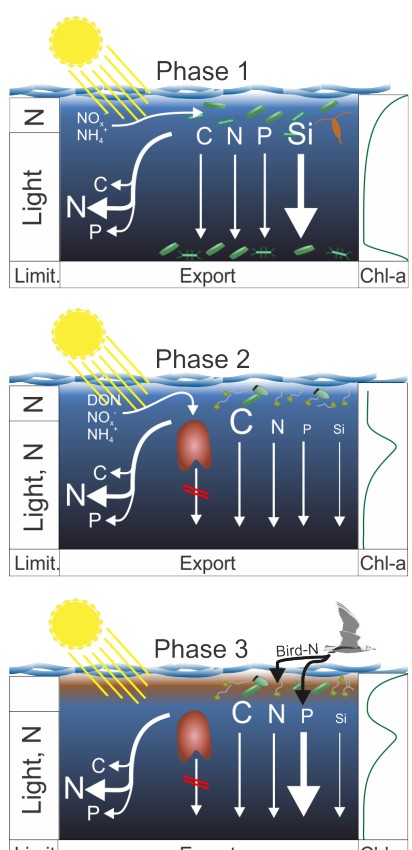


**Figure 10**. Synthesis graphic. The text in section 5 functions as an extended figure caption and
should be read to fully understand processes illustrated in this graphic. The left column indicates
the factors limiting productivity in the uppermost and the lower water column. The arrows on
the left identify which elements were remineralized preferentially during sinking. The arrows
on the right indicate the export flux of these elements. In both cases strength is indicated by the
arrow and letter sizes. The column on the right shows the approximate chl-a profile during the
three phases. The brown blob drawn in pictures of Phase 2 and 3 illustrates *A. sanguinea*.