# Peer review of "Factors controlling plankton community production, export flux, and particulate matter"

_Biogeosciences, 2020_

## Referee Comment (RC1) · Anonymous Referee #1 · 8 May 2020

Factors controlling plankton productivity, particulate matter stoichiometry, and export flux in the coastal upwelling system off Peru by LT Bach et al

General Comments The overall quality of this paper is good because it provides new insight for production processes of a very important region of the oceans. Nevertheless, there is a basic concept that needs to be carefully considered, mainly that Production is phytoplankton biomass mg C m-3 and Productivity is carbon production in mg C m-3 d-1. Please revise because biomass was measured and change it accordingly, or explain. This is a crucial point.

I think the Discussion in too long. It takes 21 pages out of 54. I cannot suggest how it can be shortened. Maybe move some ideas to the description of the results? Or make new comparative Figures for stoichiometric results?

Finally, authors refer in the Discussion in Line 677 that... there was little potential to detect treatment differences, especially in light of the large differences in the starting condition that induced considerable variance between replicates. Therefore, they decided to focus on the analyses of temporal developments of ecological and biogeochemical processes rather than on detecting treatment differences. Maybe just to make it clear, this point about variance should not be left aside and they should add a paragraph about Plankton patchiness, insufficient replicates??

Two questions are asked 1) How do plankton community structure and associated biogeochemical processes change following an upwelling event. This first question was addressed by simply monitoring the developments within the mesocosms for a 50 days' period. 2) How does upwelling of water masses with different OMZ-signatures influence plankton succession and pelagic biogeochemistry

Specific comments Abstract 1. The phytoplankton communities were initially dominated by diatoms but shifted towards a pronounced dominance of the mixotrophic harmful dinoflagellate (Akashiwo sanguinea) when inorganic nitrogen was exhausted in surface layers. It is not clear if the phrase refers to the mesocosmos enclosed waters or the natural Pacific surface layers 2. It is not clear why the increase and dominance of one dinoflagellate species is not considered a bloom? 3. Authors state that numerous biotic and abiotic factors modify productivity and biogeochemical processes. It is not clear why they simplify at the end only to nutrients and light? 4. Mesocosm study revealed key links between ecological and biogeochemical processes... Please expand and be specific here.

Introduction Eastern boundary upwelling systems (EBUS) are hotspots of marine life. References from other EBUS are missing. Please add. Moreover most self-references

for the coastal upwelling system off Peru are reiterated in lines 69 to 86. Lines 100-101 . . .the observed patterns of productivity and export in the Peruvian upwelling system (and elsewhere). . .change elsewhere at least to Perú-Chile Current or give references. Lines 103-4 . . .climate change (Add here the reference of Gruber, 2011 through warming up, turning sour, losing breath) and alterations in productivity could disrupt one of the largest fisheries in the world (maybe a different reference here).

Methods Details of very careful physical and chemical procedures are presented. Figure 2 is very unclear. Please improve Line 137 Surrounding Pacific as in line 183 should say surrounding Pacific water. Line 238 says Pacific surface waters. Please keep this nomenclature. Therefore, later in the results and Discussion the term "Pacific" would be better understood. Pacific is an Ocean so this is confusing specially referring to Redfield ratios.

Results The legend of Figures need to be homogenized. It's difficult to understand them when different denominations are used: surface and bottom waters versus uppermost and the lower water column Fig 3 The black or white lines on top of the contours show the depth integrated water column average- I don't see the white lines. Figure 4. Inorganic and organic nutrient concentrations . . .Add to legend mesocosmos in colour lines. Line in black Pacific is the control water? Very interesting to compare NO3- + NO2- and DON opposite behavior in control and mesocosmos Figure 5. Chlorophyll a concentration. . . please describe what is the black line Fig 10 Brown blob is not a very nice representation of the dinoflagellate Akashiwo. At least add the flagella

Discussion First, 4.3.1. Productivity is a rate. . . Please change to biomass production. See my General Comments. Line 1051 Altogether, our study revealed some important factors controlling plankton productivity. Authors measured Chl-a. This needs to be clarified in the whole manuscript.

Second, I don't understand why Orni-eutrophication was not included in the Abstract, I think it is a very interesting result: Orni-eutrophication during the last 10 days enabled

rapid phytoplankton growth through the relief from N-limitation and high light intensities in the uppermost meters. Bird defecation triggered intense phytoplankton blooms in most mesocosms in the uppermost part of the water column where light was plentiful. N inputs through these excrements were directly utilized and converted into organic biomass whereas the defecated P remained unutilized and sank through the water column directly into the sediment traps. Line 702 what seabirds typically add to the water column of the Pacific in this region (Otero et al., OP CIT).

Third, Export flux: I think authors should again compare their results with other sites in the HCS such as i.e. Gonzalez et al 2009, Carbon fluxes within the epipelagic zone of the Humboldt Current System off Chile: The significance of euphausiids and diatoms as key functional groups for the biological pump. Line 902.

---

## Referee Comment (RC2) · Anonymous Referee #2 · 29 May 2020

In their paper, Bach et al. describe the results from a mesocosm experiment offshore of Peru. The aim of the experiment was to compare upwelling effects on plankton communities in two different water bodies, but the authors did not achieve enough differences in biogeochemical properties of the water bodies to make any assumptions on the treatment effects as they stated in the abstract. Instead they decided to provide a descriptive manuscript and discuss the dynamic of the phytoplankton bloom over the course of the experiment.

Although I do appreciate the detail and the fairness of the description (especially in

chapter 4.1), I also think that the paper is lacking focus and would recommend some shortenings. I would recommend that the authors focus on the productivity and export processes (chapter 4.3) and leave out phytoplankton-zooplankton interactions, which are supposed to be described in detail in another paper in the special issue. Chapter 4.3 is definitely the most interesting from the scientific point of view and also well written in comparison to the other parts of the manuscript. Now the paper is extremely long and contains several stories, which do not come together. There are also 3 overall problems that I have with this paper, which require more attention:

1. Language.
The authors should carefully read the paper and get rid of the jargon and odd phrases. Some examples (there are many more in text): Second, the high primary production fuels secondary production Our paper kicks off a Biogeosciences special issue using a manual kitesurf pump so that the sediment material was sucked through the hose - first, it's a kite pump, not kitesurf pump, second, a kite pump is actually nothing else than a manual air pump (although I appreciate the authors' hobby). The water columns enclosed at the beginning of the study were temperature stratified - should be: thermally stratified Dinophyceae became about as dominant as in the other mesocosms when Cryptophyceae disappeared Nevertheless, we observed a few temporal trends that were sufficiently clear - temporal trend is something that should be statistically proven and there are methods to detect temporal trends in time series The quasi absence of silicoflagellates key mechanism muting phytoplankton growth Thus, the shift in PON:TPP in the mesocosms was triggered by ecology whereas it was arguably triggered by a physiological response in the Pacific.

2. Regime shifts. I have an impression that the authors don't entirely understand the concepts of alternative stable states and regime shifts. There are methods to detect the occurrence of a regime shift from data, but no such method has been applied. For example, "Overall, biogeochemical pools and fluxes were surprisingly constant in between the ecological regime shifts." - the authors probably mean "between alternative stable states"? I don't see any second regime shift to compare with. If the biogeochemistry was not different between the alternative stable states, how the ecological regime shift should have occurred? There are two ways of dealing with this problem: (i) carefully rewrite parts of the text that refer to the regime shifts, or (ii) perform a proper statistical analysis (plethora of methods exist from rather simple "signal to noise ratio" to more sophisticated Monte Carlo models). Personally, I don't think that we are witnessing a regime shift here, but rather a phytoplankton secession that is a result of competition.

3. Statistics. The manuscript is very descriptive in it's nature, but it does contain some statements that can and should be proven by a proper statistical test, especially considering that the title refers to "factors controlling plankton productivity" etc. What are these factors? Do they significantly affect plankton productivity? For example, the authors write on the page 18: "Nevertheless, we observed a few temporal trends that were sufficiently clear (and consistent with other datasets) so that we are confident that they were "real" and outside the noise of the measurement." - temporal trends can be determined from these data, so it should be tested if they are "sufficiently clear" (or as one might have said "significant"). Another example: "Interestingly, there was a tendency of decreasing POC:TPP during periods of chl-a increase" - it is possible to test statistically if time series are correlated (e.g. using cross-correlograms)

Other comments: "The nutrient concentration between the collected water bodies were relatively small" - the authors decided to exclude treatment effects from the analysis based on this assumption, but in the results section they report that OMZ source water at the station 3 contained 4 umol/L NOx and from station 1 only 0.3 umol/L NOx. Is this a small difference? In fact, the authors describe that low concentrations of NOx in the mesocosms with the OMZ from the station 1 lead to the decrease of the NOx concentration following the water addition. As the OMZ water contained 2.5 umol/L PO4 at both stations, the treatments must have differed in N:P ratios after OMZ water addition. If this is the case, I would argue that the authors need more solid arguments to ignore the treatment effects than simply their personal judgement. One possibility is

to statistically prove that the treatments did not affect water chemistry.

Line 527: diatoms - change into Bacillariophyceae for consistency.

The authors discuss the results from flow cytometry and microscopy. These methods should be described in the method section. Also grazing experiment of Paracalanus is not described in the method section, but discussed later in text.

Line 717: these principles come back to the papers by Margalef and Reynolds, which would be proper citations

Line 724: migratory - change into "motile" Paragraph from the line 792 onwards is full of jargon and describes the effects of enclosure rather than sampling.

Line 808: physical processes - change into "transport processes" Paragraph from 873 onwards can be omitted

Line 898: it's not a surprise, dinoflagellates like Akashiwo and Alexandrium often have a long lag phase.

Line 969: it is not contradictory to this study. Hillebrand et al. is based on species specific population growth rates (u) and not on communities.

Lines 1000-1002: this is important for interpretation and should come much earlier in the manuscript, when you describe the mesocosms and experimental design.

Paragraph from the line 1042 can be omitted, it doesn't bring anything new to this study

---

## Author Comment (AC1) · 30 Jun 2020

We thank both reviewers for their insightful comments, which helped to improve the manuscript. Please find our point by point responses in the following. Please note that line numbers in our responses refer to the revised version of the manuscript.

Reviewer #1

1) General Comments: The overall quality of this paper is good because it provides new insight for production processes of a very important region of the oceans. Nevertheless, there is a basic concept that needs to be carefully considered, mainly that Production is phytoplankton biomass mg C m-3 and Productivity is carbon production in mg C m-3 d-1. Please revise because biomass was measured and change it accordingly, or explain. This is a crucial point.

REPLY: We thank the reviewer for the kind words. We changed "productivity" to "production" throughout the manuscript.

2) I think the Discussion in too long. It takes 21 pages out of 54. I cannot suggest how it can be shortened. Maybe move some ideas to the description of the results? Or make new comparative Figures for stoichiometric results?

REPLY: We thank the reviewer for this suggestion, which is consistent with the feedback we got from reviewer #2. We therefore removed section 4.2 ("plankton succession") from the discussion. This aspect will be covered in more specialized manuscripts within this special issue. Furthermore, zooplankton results, methodology, and the zooplankton figure were removed as this aspect was not covered in the discussion any longer. We also deleted more speculative parts of the discussion in sections 4.3 and 4.4. The discussion is considerably shorter in the revised version of the manuscript and more focussed on the biogeochemistry.

3) Finally, authors refer in the Discussion in Line 677 that. . . there was little potential to detect treatment differences, especially in light of the large differences in the starting condition that induced considerable variance between replicates. Therefore, they decided to focus on the analyses of temporal developments of ecological and biogeochemical processes rather than on detecting treatment differences. Maybe just to make it clear, this point about variance should not be left aside and they should add a paragraph about Plankton patchiness, insufficient replicates??

REPLY: We revised the mentioned text to indicate that a higher number of replicates may have helped to detect treatment differences due to increased statistical power. However, we also noted that this is unfeasible due to the enormous costs of in situ

mesocosm experimentation.

4) Abstract 1. The phytoplankton communities were initially dominated by diatoms but shifted towards a pronounced dominance of the mixotrophic harmful dinoflagellate (Akashiwo sanguinea) when inorganic nitrogen was exhausted in surface layers. It is not clear if the phrase refers to the mesocosmos enclosed waters or the natural Pacific surface layers.

REPLY: We added the word "mesocosm" to clarify (Line 52).

5) Abstract 2. It is not clear why the increase and dominance of one dinoflagellate species is not considered a bloom?

REPLY: We added the word "bloom" to indicate that A. sanguinea formed a bloom (Line 56).

6) Abstract 3. Authors state that numerous biotic and abiotic factors modify productivity and biogeochemical processes. It is not clear why they simplify at the end only to nutrients and light?

REPLY: The reasons why we focussed on light and nutrients are described and justified in the discussion line 687-699. This description would be too long for the abstract but we think there are no open questions in the abstract at this point of the abstract, since we emphasize nutrients and light but do (at this point) not exclude any others.

7) Abstract 4. Mesocosm study revealed key links between ecological and biogeochemical processes. . . Please expand and be specific here.

REPLY: This sentence is to close the abstract and refers to the observations described in sentences before. We modified this sentence slightly to link it better to the previous text.

8) Introduction 1. Eastern boundary upwelling systems (EBUS) are hotspots of marine life. References from other EBUS are missing. Please add.

REPLY: The cited reference is a review of all 4 major EBUS (North-west Africa, Benguela, Humboldt, California). So we think that it is an appropriate reference for our statement. Nevertheless, we also refer to the review by (Thiel et al., 2007) in the revised version to add a reference specific to the Humboldt system.

9) Introduction 2. Moreover most self-references for the coastal upwelling system off Peru are reiterated in lines 69 to 86. Lines 100-101 ...the observed patterns of productivity and export in the Peruvian upwelling system (and elsewhere). . .change elsewhere at least to PeruÌA̧-Chile Current or give references.

REPLY: The reviewer asks for more "diversity" with respect to references that support our more general statements. To account for this request, we added a reference by (Bakun and Weeks, 2008; Daneri et al., 2000; González et al., 2009; Thiel et al., 2007). As our study region is in Peru, we think it is well justified to refer to the widely used term "Peruvian upwelling" in our study. We think this is more appropriate for our case than e.g. Humboldt system, since using the term "Peruvian upwelling" reminds the reader where our study took place.

10) Introduction 3. Lines 103-4 . . .climate change (Add here the reference of Gruber, 2011 through warm- ing up, turning sour, losing breath) and alterations in productivity could disrupt one of the largest fisheries in the world (maybe a different reference here).

REPLY: We moved the Gruber reference to the previous part as suggested by the reviewer. The second statement is now supported by referring to Bakun and Weeks, 2008 (Lines 101-104).

11) Methods 1. Figure 2 is very unclear. Please improve

REPLY: We simplified the figure and added frames to better distinguish the individual subplots.

12) Surrounding Pacific as in line 183 should say surrounding Pacific water. Line 238 says Pacific surface waters. Please keep this nomenclature. Therefore, later in the

results and Discussion the term "Pacific" would be better understood. Pacific is an Ocean so this is confusing specially referring to Redfield ratios.

REPLY: We changed "Pacific" to "Pacific water" throughout the text as suggested by the reviewer.

13) Results 1. The legend of Figures needs to be homogenized. It's difficult to understand them when different denominations are used: surface and bottom waters versus uppermost and the lower water column

REPLY: We changed the term "uppermost" to "surface" and specified the depth range when necessary.

14) Fig 3 The black or white lines on top of the contours show the depth integrated water column average- I don't see the white lines.

REPLY: We thank the reviewer for pointing this out. There are either black or white lines for the water column average. The "color" differs due to visibility reasons. We indicated that black lines are used in subplots A, B, D and white lines in suplot C.

15) Figure 4. Inorganic and organic nutrient concentrations . . .Add to legend mesocosmos in colour lines. Line in black Pacific is the control water?

REPLY: The legend is shown in subplot G. The legend also indicates that the black line represents data from the surrounding Pacific water.

16) Very interesting to compare NO3- + NO2- and DON opposite behavior in control and mesocosmos REPLY: The differences are due to the differences in the sampling strategy. While the mesocosms are a lagrangian system (same water mass sampled every other day), the Pacific water changes from sampling day to sampling day due to advective processes. Thus, temporal trends in the two systems cannot really be compared.

17) Figure 5. Chlorophyll a concentration. . . please describe what is the black line

REPLY: We thank the reviewer for pointing towards this. The black lines were removed from the plots as they were in there by mistake.

18) Fig 10 Brown blob is not a very nice representation of the dinoflagellate Akashiwo. At least add the flagella

REPLY: We added flagella and changed the shape of Akashiwo to more closely resemble its natural appearance.

19) Discussion First, 4.3.1. Productivity is a rate. . . Please change to biomass production. See my General Comments. Line 1051 Altogether, our study revealed some important factors controlling plankton productivity. Authors measured Chl-a. This needs to be clarified in the whole manuscript.

REPLY: We changed productivity to production throughout the text as suggested by the reviewer.

20) Second, I don't understand why Orni-eutrophication was not included in the Abstract, I think it is a very interesting result: Orni-eutrophication during the last 10 days enabled rapid phytoplankton growth through the relief from N-limitation and high light intensities in the uppermost meters. Bird defecation triggered intense phytoplankton blooms in most mesocosms in the uppermost part of the water column where light was plentiful. N inputs through these excrements were directly utilized and converted into organic biomass whereas the defecated P remained unutilized and sank through the water column directly into the sediment traps. Line 702 what seabirds typically add to the water column of the Pacific in this region (Otero et al.).

REPLY: We did not pick up on orni-eutrophication in the abstract as this effect is amplified profoundly through the mesocosms (because the birds had a structure to sit on). Thus, it is an artificial effect that helped us to understand some processes but the orni-eutrophication itself is not a key outcome of our study.

21) Third, Export flux: I think authors should again compare their results with other

sites in the HCS such as i.e. Gonzalez et al 2009, Carbon fluxes within the epipelagic zone of the Humboldt Current System off Chile: The significance of euphausiids and diatoms as key functional groups for the biological pump. Line 902.

REPLY: We thank the reviewer for this suggestion. The comparison of vertical fluxes in the mesocosms with those at the surrounding Pacific will be addressed in a specific paper in this special issue by Ursula Mendoza et al. They had sediment traps installed next to the mesocosms in the Pacific and will investigate in- and outside fluxes and compare them with measurements from other regions. However, we refer to Gonzalez et al. 2009 in the introduction as it is a relevant paper for our study.

References

Bakun, A. and Weeks, S. J.: The marine ecosystem off Peru: What are the secrets of its fishery productivity and what might its future hold?, Prog. Oceanogr., 79(2–4), 290–299, doi:10.1016/j.pocean.2008.10.027, 2008.

Daneri, G., Dellarossa, V., Quiñones, R., Jacob, B., Montero, P. and Ulloa, O.: Primary production and community respiration in the Humboldt Current System off Chile and associated oceanic areas, Mar. Ecol. Prog. Ser., 197, 41–49, doi:10.3354/meps197041, 2000.

González, H. E., Daneri, G., Iriarte, J. L., Yannicelli, B., Menschel, E., Barría, C., Pantoja, S. and Lizárraga, L.: Carbon fluxes within the epipelagic zone of the Humboldt Current System off Chile: The significance of euphausiids and diatoms as key functional groups for the biological pump, Prog. Oceanogr., 83(1–4), 217–227, doi:10.1016/j.pocean.2009.07.036, 2009.

Thiel, M., Macaya, E. C., Acuña, E., Arntz, W. E., Bastias, H., Brokordt, K., Camus, P. A., Castilla, J. C., Castro, L. R., Cortés, M., Dumont, C. P., Escribano, R., Fernández, M., Gajardo, J. A., Gaymer, C. F., Gomez, I., González, A. E., González, H. E., Haye, P. A., Illanes, J.-E., Iriarte, J. L., Lancellotti, D. A., Luna-Jorquera, G., Luxoro, C., Manríquez, P. H., Marín, V., Muñoz, P., Navarrete, S. A., Perez, E., Poulin, E., Sellanes, J., Sepúlveda, H. H., Stotz, W., Tala, F., Thomas, A., Vargas, C. A., Vasquez, J. A. and Alonso Vega, J. .: the Humboldt Current System of Northern and Central Chile Oceanographic Processes Ecological Interactions and Socioeconomic Feedback, Oceanogr. Mar. Biol. An Annu. Rev., 45, 195–344, 2007.

———————————————————

---

## Author Comment (AC2) · 30 Jun 2020

We thank both reviewers for their insightful comments, which helped to improve the manuscript. Please find our point by point responses in the following. Please note that line numbers in our responses refer to the revised version of the manuscript.

22) In their paper, Bach et al. describe the results from a mesocosm experiment off-shore of Peru. The aim of the experiment was to compare upwelling effects on plankton communities in two different water bodies, but the authors did not achieve enough differences in biogeochemical properties of the water bodies to make any assumptions on the treatment effects as they stated in the abstract. Instead they decided to provide a descriptive manuscript and discuss the dynamic of the phytoplankton bloom over the course of the experiment. Although I do appreciate the detail and the fairness of the description (especially in chapter 4.1), I also think that the paper is lacking focus and would recommend some shortenings. I would recommend that the authors focus on the productivity and export processes (chapter 4.3) and leave out phytoplankton-zooplankton interactions, which are supposed to be described in detail in another paper in the special issue. Chapter 4.3 is definitely the most interesting from the scientific point of view and also well written in comparison to the other parts of the manuscript. Now the paper is extremely long and contains several stories, which do not come together.

REPLY: We thank the reviewer for the valuable comments. We removed section 4.2 (Plankton succession section) and section 4.5 as suggested by reviewer #2. We kept section 4.4 (C:N:P:Si stoichiometry in the mesocosms) since stoichiometry is an interesting parameter to discuss and is not covered in another manuscript of the special issue. The discussion is now more focussed and shorter.

23) Language 1. The authors should carefully read the paper and get rid of the jargon and odd phrases. Some examples (there are many more in text): Second, the high primary production fuels secondary production.

REPLY: Changed to "...it sustains one of the largest fisheries in the world, making the Peruvian upwelling system an area of outstanding economic value." (Lines 77-79)

24) Language 2. Our paper kicks off a Biogeosciences special issue.

REPLY: Changed to: "Our paper is the first in a Biogeosciences special issue about the 2017 Peru mesocosm campaign." (Lines 116-117).

25) Language 3. using a manual kitesurf pump so that the sediment material was

sucked through the hose - first, it's a kite pump, not kitesurf pump, second, a kite pump is actually nothing else than a manual air pump (although I appreciate the authors' hobby).

REPLY: It is a fantastic hobby indeed :-D We changed this to "air pump". (Line 242)

26) Language 4. The water columns enclosed at the beginning of the study were temperature stratified - should be: thermally stratified

REPLY: we changed temperature to "thermally" (Line 379).

27) Language 5. Dinophyceae became about as dominant as in the other mesocosms when Cryptophyceae disappeared

REPLY: We changed this part to: "The A. sanguinea bloom was delayed by ∼10 days in M3 and they remained absent in M4 throughout the study. Cryptophypheae benefited from the absence of A. sanguinea and were the dominant group in M3 and M4 in the ∼10 days after the OMZ water addition" (Line 351-354).

28) Language 6. Nevertheless, we observed a few temporal trends that were sufficiently clear - temporal trend is something that should be statistically proven and there are methods to detect temporal trends in time series

REPLY: This sentence was deleted.

29) Language 7. The quasi absence of silicoflagellates

REPLY: This sentence was deleted.

30) Language 8. key mechanism muting phytoplankton growth

REPLY: Agreed, this part was weird. We changed it to: "It appears that self-shading due to high biomass is a key mechanism that constrains phytoplankton growth when integrated over the water column. This constraint may enable an equilibrium between production and loss processes as reflected in the relative constancy of chl-a, POCWC

and POCST (Figs. 5A and 8A, E; see next section for further details on export). Indeed, the orni-eutrophication demonstrates that when limiting nutrients are added to a layer with high light intensity, phytoplankton can break this equilibrium and grow rapidly (Fig. 5A)." (Line 742-748)

31) Thus, the shift in PON:TPP in the mesocosms was triggered by ecology whereas it was arguably triggered by a physiological response in the Pacific.

REPLY: This section was deleted to shorten the manuscript.

32) Regime shifts. I have an impression that the authors don't entirely understand the concepts of alternative stable states and regime shifts. There are methods to detect the occurrence of a regime shift from data, but no such method has been applied. For example, "Overall, biogeochemical pools and fluxes were surprisingly constant in between the ecological regime shifts." - the authors probably mean "between alternative stable states"? I don't see any second regime shift to compare with. If the biogeochemistry was not different between the alternative stable states, how the ecological regime shift should have occurred? There are two ways of dealing with this problem: (i) carefully rewrite parts of the text that refer to the regime shifts, or (ii) perform a proper statistical analysis (plethora of methods exist from rather simple "signal to noise ratio" to more sophisticated Monte Carlo models). Personally, I don't think that we are witnessing a regime shift here, but rather a phytoplankton secession that is a result of competition.

REPLY: Agreed. We do not refer to regime shifts in the revised version. Instead, we refer to changes in community composition.

33) Statistics. The manuscript is very descriptive in it's nature, but it does contain some statements that can and should be proven by a proper statistical test, especially considering that the title refers to "factors controlling plankton productivity" etc. What are these factors? Do they significantly affect plankton productivity? For example, the authors write on the page 18: "Nevertheless, we observed a few temporal trends that

were sufficiently clear (and consistent with other datasets) so that we are confident that they were "real" and outside the noise of the measurement." - temporal trends can be determined from these data, so it should be tested if they are "sufficiently clear" (or as one might have said "significant"). Another example: "Interestingly, there was a tendency of decreasing POC:TPP during periods of chl-a increase" - it is possible to test statistically if time series are correlated (e.g. using cross-correlograms)

REPLY: The sentences the reviewer is referring to (and similarly vague sentences) were removed from the revised version of the manuscript. In general, it is important to emphasize that we determined many corresponding parameters in our time-series. This allows us to confirm a trend in a certain parameter by checking for consistency with related parameters. For example, in section 4.2.1 (formerly 4.3.1) we describe a strong POC increase and argue that this increase is "real" (i.e. significant), as it coincides with an increase of PON, POP, Chla, dinoflagellate abundance and a decrease in DIC, PO43-, DON. We argue that these "mechanistic" insights are more powerful in confirming/rejecting a trend than statistical approaches, which are often based on certain choices (see below). We carefully explored the possibility to use statistical tools for the detection of temporal trends (and in fact use some of them like moving averages for data exploration). One option is to use regression analyses. However, the outcome (i.e. significant or not significant) depends on choice of the applied regression model (linear or non-linear) and on the segment of the time-series that is explored (e.g. in the example above the POC increase due to the dinoflagellate bloom takes place over ~10 days (Fig. S1) but it is difficult to clearly determine the exact time frame due to limited data on dinoflagellate abundances). These choices include a certain level of arbitrariness, which can influence the outcome. Thus, we generally put more trust into a "mechanistic" explanation of an observed trend. As suggested by the reviewer, we also explored the possibility to apply cross-correlograms to detect trends. However, this tool is used to detect periodicities in the time-series which may be hidden by the noise of the measurements. Periodicities can be caused e.g. by tidal, diel, lunar, seasonal cycles etc. None of these external forcings applied to our time-series as tidal shifts are

excluded in mesocosms, the temporal resolution was too low to detect diel cycles, and the experiment too short for lunar/seasonal cycles. Nevertheless, we checked our data for periodic auto-correlations with the acf function in R, but could not find significant autocorrelations in the dataset for time-lags longer than 1 or 2 sampling days (sometimes also over longer timescales when there was very little change in a measured parameter from sampling day to sampling day (e.g. PO43-)).

34) Other comments: "The nutrient concentration between the collected water bodies were relatively small" - the authors decided to exclude treatment effects from the analysis based on this assumption, but in the results section they report that OMZ source water at the station 3 contained 4 umol/L NOx and from station 1 only 0.3 umol/L NOx. Is this a small difference? In fact, the authors describe that low concentrations of NOx in the mesocosms with the OMZ from the station 1 lead to the decrease of the NOx concentration following the water addition. As the OMZ water contained 2.5 umol/L PO4 at both stations, the treatments must have differed in N:P ratios after OMZ water addition. If this is the case, I would argue that the authors need more solid arguments to ignore the treatment effects than simply their personal judgement. One possibility is to statistically prove that the treatments did not affect water chemistry.

REPLY: The source water had the above-mentioned NOx concentrations but it was diluted afterwards when mixed with the mesocosm water (see Section 2.3). We added a table to the revised version that summarizes treatment-specific information (such as NOx addition, Table 1). NOx-, NH4+, and N:P were significantly different between the treatments after the OMZ water addition (this information was added to section 3.2 and Table 1). However, differences were small (e.g. N/P =1.5 vs. 2.9) and the variance among the mesocosms was large due to variable initial conditions (as discussed in section 4.1). We may still be able to reveal some significant differences between the two treatments in some parameters (e.g. chl-a) when applying more sophisticated statistics but our impression was that we can learn more from this dataset when focussing on temporal developments. Our decision not to discuss treatment differences
has no influence on the conclusions drawn in this manuscript. This is because we are focussing on temporal changes within individual mesocosms and also interpret the results individually when trends in mesocosms differ from each other. It is also important to note that some manuscripts to be published within this special issue will have a look at treatment differences (we emphasized this in the revised version of the manuscript) (Line 650-652).

35) Line 527: diatoms - change into Bacillariophyceae for consistency.

REPLY: We changed diatoms to Bacillariophyceae. (Line 521)

36) The authors discuss the results from flow cytometry and microscopy. These methods should be described in the method section. Also grazing experiment of Paracalanus is not described in the method section, but discussed later in text.

REPLY: The zooplankton grazing experiments are not discussed anymore in the manuscript (all zooplankton data was removed from the manuscript). We only refer once to the imaging flow cytometry and microscopy datasets in the discussion to provide the information that the Dinophyceae dominance (estimated with CHEMTAX) is due to the bloom of Akashiwo sanguineum. The flow cytometry/ microscopy datasets will be described/discussed in detail by Avy Bernales et al. in a specialized phytoplankton paper so it may be unnecessary to add a full section on flow cytometry and microscopy only to provide this information.

37) Line 717: these principles come back to the papers by Margalef and Reynolds, which would be proper citations

REPLY: Section 4.2 was deleted.

38) Line 724: migratory - change into "motile" Paragraph from the line 792 onwards is full of jargon and describes the effects of enclosure rather than sampling.

REPLY: Section 4.2 was deleted.

39) Line 808: physical processes - change into "transport processes" Paragraph from 873 onwards can be omitted

REPLY: We changed "physical processes" to "transport processes" (Lines 687-690) and deleted the mentioned paragraph as suggested by the reviewer.

40) Line 898: it's not a surprise, dinoflagellates like Akashiwo and Alexandrium often have a long lag phase.

REPLY: We agree that it is probably not surprising for plankton ecologists but there is debate around this topic in the "biological pump" community (see the referenced articles by (Laws and Maiti, 2019; Stange et al., 2017)). Therefore, the information is relevant in the context of export fluxes.

41) Line 969: it is not contradictory to this study. Hillebrand et al. is based on species specific population growth rates (u) and not on communities.

REPLY: This paragraph was deleted.

42) Lines 1000-1002: this is important for interpretation and should come much earlier in the manuscript, when you describe the mesocosms and experimental design.

REPLY: The classification of the experiment into distinct phases 1-3 is based upon combining all information discussed in this manuscript. These phases were not designed beforehand so we think this classification needs to go into the synthesis section 5.

43) Paragraph from the line 1042 can be omitted, it doesn't bring anything new to this study

REPLY: We deleted this paragraph as suggested by the reviewer.

References

Laws, E. A. and Maiti, K.: The relationship between primary production and export

production in the ocean: Effects of time lags and temporal variability, Deep Sea Res. Part I Oceanogr. Res. Pap., doi:10.1016/j.dsr.2019.05.006, 2019.

Stange, P., Bach, L. T., Le Moigne, F. A. C., Taucher, J., Boxhammer, T. and Riebesell, U.: Quantifying the time lag between organic matter production and export in the surface ocean: Implications for estimates of export efficiency, Geophys. Res. Lett., 44(1), 268–276, doi:10.1002/2016GL070875, 2017.